# A Comprehensive Study of Real-Time Object Detection Networks Across Multiple Domains: A Survey

**Elahe Arani**[‡1,2]**, Shruthi Gowda**[‡1]**, Ratnajit Mukherjee**[1]**, Omar Magdy**[1]**, Senthilkumar Kathiresan**[1]**, Bahram Zonooz**[1,2]

*{elahe.arani, shruthi.gowda}@navinfo.eu, bahram.zonooz@gmail.com*
[1] *Advanced Research Lab, NavInfo Europe, Eindhoven, The Netherlands*
[2] *Department of Mathematics and Computer Science, Eindhoven University of Technology, The Netherlands*
[‡] *Contributed equally.*

**Reviewed on OpenReview:** *https://openreview.net/forum?id=ywr5sWqQt4*

## Abstract

Deep neural network based object detectors are continuously evolving and are used in a multitude of applications, each having its own set of requirements. While safety-critical applications need high accuracy and reliability, low-latency tasks need resource and energy-efficient networks. Real-time detection networks, which are a necessity in high-impact real-world applications, are continuously proposed but they overemphasize the improvements in accuracy and speed while other capabilities such as versatility, robustness, resource, and energy efficiency are omitted. A reference benchmark for existing networks does not exist nor does a standard evaluation guideline for designing new networks, which results in ambiguous and inconsistent comparisons. We, therefore, conduct a comprehensive study on multiple real-time detection networks (anchor-based, keypoint-based, and transformer-based) on a wide range of datasets and report results on an extensive set of metrics. We also study the impact of variables such as image size, anchor dimensions, confidence thresholds, and architecture layers on the overall performance. We analyze the robustness of detection networks against distribution shift, natural corruptions, and adversarial attacks. Also, we provide the calibration analysis to gauge the reliability of the predictions. Finally, to highlight the real-world impact, we conduct two unique case studies, on autonomous driving and healthcare application. To further gauge the capability of networks in critical real-time applications, we report the performance after deploying the detection networks on edge devices. Our extensive empirical study can act as a guideline for the industrial community to make an informed choice on the existing networks. We also hope to inspire the research community towards a new direction of design and evaluation of networks that focuses on the bigger and holistic overview for a far-reaching impact.

## 1 Introduction

Recent advancements in deep neural networks have led to remarkable breakthroughs in the field of object detection. Object detection combines both classification and localization by providing the locations of the object instances along with the class label and confidence scores. Detectors find their way into multiple applications such as autonomous driving systems (ADS), surveillance, robotics, and healthcare. An ADS needs to detect vehicles, traffic signs, and other obstructions accurately in real-time, and additionally, to ensure safety, they need the detectors to perform reliably and consistently in different lighting and weather conditions. Healthcare applications require high accuracy even if it is not extremely fast. Low-latency applications require deployment on edge devices and hence need detectors to be fast and also compact enough to fit on low-power hardware devices. Different applications have different criteria and real-world settings come with time and resource constraints. Therefore, detectors need to be resource and energy

Table 1: The summary of all the detection heads, backbones, datasets, and deployment hardware used for the experiments in this study.

| Detection Head | Backbone | Dataset | Hardware |
|---|---|---|---|
| ThunderNet | ShuffleNet-v2; EfficientNet-B0 | VOC; COCO | NVIDIA 2080Ti |
| YOLO; SSD | MobileNet-v2; DeiT-T | Corrupted COCO | Jetson Xavier |
| DETR | DarkNet-19; ResNet-18 | BDD; Cityscapes | Jetson TX2 |
| CenterNet; TTFNet; FCOS; NanoDet | Xception; HarDNet-68; VoVNet | Kvasir-SEG | |

efficient to ensure deployment in high-impact real-world applications. This calls for a detailed analysis of real-time detection networks on different criteria.

A number of real-time detection networks have been proposed which achieve state-of-the-art performance and while they primarily focus on reporting accuracy and speed, other metrics such as simplicity, versatility, and energy efficiency are omitted. As newer detectors are being proposed frequently, without a standard evaluation framework, the trend is moving more towards a short-sighted direction that focuses only on nominal improvements. Furthermore, there is ambiguity in the parameters used which makes reproducibility and fair comparison hard. Underplaying metrics such as resource and energy consumption can also result in a negative environmental impact (Badar et al., 2021). Several survey papers (Liu et al., 2018a; Zhao et al., 2018b; Zou et al., 2019) provide an overview of detection networks over the years but certain gaps prevent them from being used as a standard benchmark. Most of the analyses are done on a subset of networks (mostly on older two-stage architectures) and do not focus on real-time detectors. Hence, this won't prove beneficial in comparing and choosing networks for real-time applications. Also, the focus is on accuracy and speed but the network capabilities such as generalization, robustness, and reliability are not covered. The studies are limited to a few networks and datasets and the results are reported on desktop GPU only. These shortcomings present a challenge in comparing existing detection networks in a fair and detailed manner to choose a network for a specific application, and also in designing new object detectors as there is neither a detailed reference benchmark nor a standard evaluation guideline.

We, therefore, conduct a comprehensive study on real-time object detection on multiple detectors on diverse datasets along with the genetic object detection benchmarks, we also provide two case studies on autonomous driving and healthcare applications. In addition to accuracy and inference time, we evaluate the resource and energy consumption of each model to estimate the environmental impact. We choose a multitude of networks and create a uniform framework where a different combination of backbones (or feature extractors) and detection heads can be analyzed with ease (Table 1). To further assess the performance gain/loss in detail, we decouple the effects of different variables such as image size, anchor size, confidence thresholds, and architecture layer types. For a uniform evaluation, we follow a standard experimentation pipeline and report all the parameters used. Each combination is trained and tested on two widely used generic datasets (PASCAL VOC (Everingham et al., 2010) and MS COCO (Lin et al., 2014)). Networks should be robust to changing lighting and weather conditions in real-time applications, hence, we further do an extensive robustness analysis to analyze the results of the networks on distribution shift and natural corruptions. For safety-critical applications, networks should be also robust to adversarial images which contain the imperceptible changes to the human's eyes, hence we evaluate the robustness of the networks to such attacks. Likewise, for these application, a measure of uncertainty is useful to take timely decisions, and hence we also provide the reliability analysis of each of the networks. Finally, to demonstrate the real-world impact, we conduct two exclusive case studies on autonomous driving and healthcare domains. For the former, we report the detector performances on the Berkeley Deep Drive (BDD) (Yu et al., 2018) dataset, which is more relevant for ADS application. We also show the generalization capability and report performance on an out-of-distribution (OOD) dataset, Cityscapes (Cordts et al., 2016). To highlight the feasibility of real-time deployment of the detectors, we deploy the NVIDIA TensorRT optimized models on embedded hardware and report the real-time performances on low-power devices. For the healthcare case study, we present the capability of the networks to detect polyps from medical images, that are used to detect cancer in patients. These applications cover two diverse domains which have different requirements and our case studies offer a unique perspective to look beyond the standard benchmarks and gauge the capability of the detectors on diverse datasets that are more relevant and applicable in real-time applications.

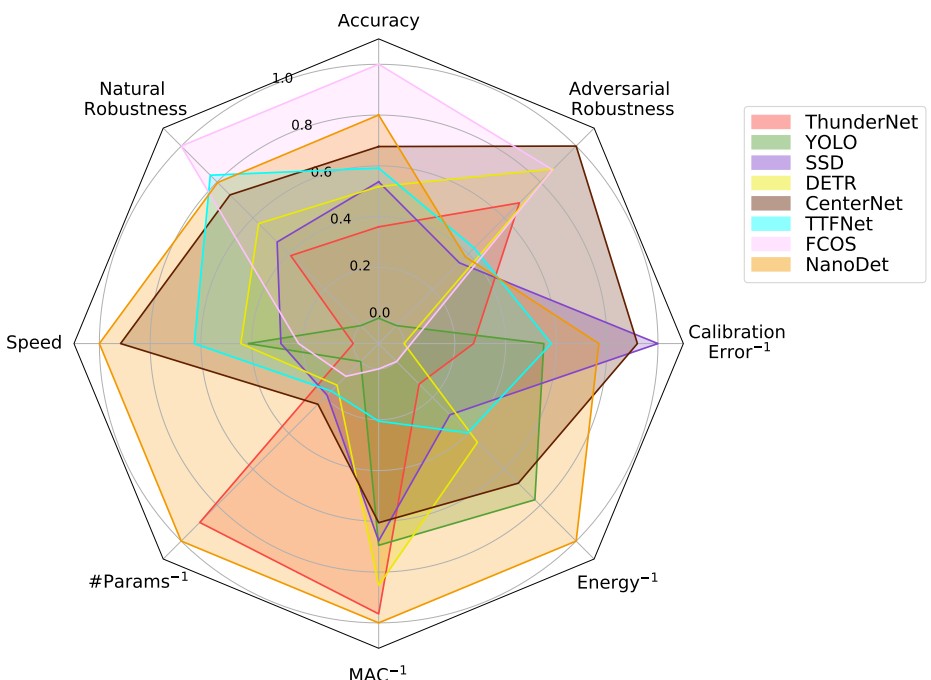

Figure 1: Overall performance of different object detector networks (with HarDNet-68 backbone) trained on COCO dataset on different metrics. All metrics are normalized w.r.t minimum and maximum while for the four metrics with $-1$ superscript, normalization was applied on the inverse values. Therefore, a network with full coverage of the octagon has all the ideal features: highest accuracy, natural/adversarial robustness, and speed with lowest number of parameters, MAC count, energy consumption, and calibration error (check Section A.1 for more details).

Our comprehensive analyses are summarized in Figure 1 where each vertex corresponds to a metric and the eight different colors represent different detectors (Check Section A.1 for more details). We report eight metrics namely, accuracy, robustness to natural and adversarial corruptions, speed, number of parameters, MAC (Multiply-Accumulate operations) count, energy consumption, and calibration error (that measures reliability). The plot is represented such that the ideal network should occupy the whole octagon and such a network has the highest accuracy, robustness, and speed, the lowest number of parameters and MAC count while consumes the lowest energy and is the best calibrated. The only real-time two-stage detector *ThunderNet*, designed for mobile devices, is efficient in resources, but falls short in accuracy, natural robustness, and is one of the slowest networks. *YOLO*, an anchor-based detector, performs second-best in energy consumption and lies in the middle range of calibration but falls behind in speed, accuracy, and robustness. *SSD*, another anchor-based detector, lies in the middle spectrum and provides a good balance between accuracy and speed. It also has the best calibration score and is more reliable. *DETR*, which is the Transformer-based detector, is the second best in terms of adversarial robustness but has the lowest calibration score and hence less reliable predictions. *CenterNet* has the highest robustness to adversarial attacks, is the second fastest, and also lies in the good spectrum on all other metrics. *TTFNet* lies in the middle spectrum. *FCOS* has the highest accuracy and robustness but falters in other metrics. *NanoDet* is the fastest and second-best in terms of accuracy, and has the lowest resource consumption. These four belong to the anchor-free keypoint-based detector category. Overall, *NanoDet* reaches the highest point on the most of the vertices and has average scores on calibration, and hence, is a contender for applications that need to be run on low-power devices with high speed and accuracy.

Apart from these extensive analyses, we also extract useful insights and details. The keypoint-based methods generally outperform the anchor-based and two-stage methods in terms of accuracy and speed. We also note that while higher MAC counts may result in higher energy consumption, they do not necessary lead to an improvement in accuracy. All the detectors provide higher accuracy on medium and large objects but

suffer in the case of small object detection. *FCOS* relatively performs better in detecting small objects when paired with heavier backbones such as *HarDNet-68*. We observe that increase in input image size is not always beneficial as the decline in speed often outnumbers the increase in accuracy. The non-deterministic way the anchor sizes affect the predictions of anchor-based detectors makes them difficult to adapt to newer datasets. Keypoint-based detectors tend to generalize well across multiple datasets. The variation in accuracy and speed seen with different confidence thresholds shows the ambiguity in reproducing the results. As transformers employ attention blocks to capture global context, they are less sensitive to varying image sizes and offer consistent performance. The calibration results show that keypoint-based methods are cautious and underconfident thus proving useful in safety-critical applications. But, interestingly, *DETR* is the most overconfident among all the networks. As transformers are gaining more traction, such detailed analysis will offer more insights into the capabilities and pitfalls of this new architectural paradigm. The case study on ADS reveals that the performance trend seen on one device does not necessarily translate to embedded hardware used in deployment. The healthcare case study shows that networks with relatively higher precision may not have higher recall values which are more important in medical data (as a false negative is more harmful than a false positive).

Overall, we highlight the importance of a standard evaluation guideline that can address ambiguous comparisons and omissions of important metrics in the present literature. To that end, we provide a holistic view of real-time detection by providing both the big picture with a more zoomed-in view into the architectures and the analyses. The industrial community can benefit by referring to the extensive results to make an informed choice about the detector based on the application requirements. We also hope to encourage the research community to focus more on essential and far-reaching issues instead of nominal performance improvements. And since new detection networks are being introduced frequently, this study can be used as a precept while designing new detectors.

All the networks (detection head and backbones), datasets, and hardware used in this study are summarized in Table 1. The contributions of the paper are summarized below:

- An extensive empirical study across combinations of nine feature extraction networks and eight detection heads ranging from two-stage, single-stage, anchor-based, keypoint-based, and transformer-based architectures.

- Detailed results including accuracy, speed, number of learnable parameters, MAC count, and energy consumption on the benchmark datasets.

- Effect of variables such as image size, anchor size, confidence thresholds, and specific architecture design layouts on the overall performance.

- Robustness analysis of all the networks against fifteen different natural corruptions and adversarial attacks with varying strength.

- Reliability analysis by assessing the calibration score of all the networks.

- A case study on Autonomous Driving Systems by performing analysis on the more relevant BDD dataset. And, generalization performance on out-of-distribution data by testing the networks on the Cityscapes dataset.

- Deployment of TensorRT-optimized detectors on edge devices: Jetson-Xavier and Jetson-Tx2.

- A case study on healthcare application by performing analysis on the Kvasir-SEG dataset to detect cancerous polyps.

## 2 Related Work

Logothetis & Sheinberg (1996) conducted one of the initial surveys approaching visual object recognition from a neuroscience perspective. Grauman & Leibe (2011) and Andreopoulos & Tsotsos (2013) conducted surveys on computer vision-based visual object recognition systems through the years. However, these surveys

focus entirely on classical computer vision techniques based on geometric properties and textures covering a broader topic of object recognition and not specifically detection. Given the rapid progress of object detection in recent years, more surveys (Liu et al., 2018a; Zhao et al., 2018b; Zou et al., 2019) have been published providing broad coverage of the deep learning-based detection techniques. Huang et al. (2017) benchmarked three object detectors along with three backbones on the COCO dataset. The paper reports accuracy and speed for Faster R-CNN (Ren et al., 2015), R-FCN (Dai et al., 2016), and SSD (Liu et al., 2016) detectors in combination with MobileNet (Howard et al., 2017), ResNet-101 (He et al., 2016), and InceptionResNet-V2 (Szegedy et al., 2016a) backbones. They mostly focus on two-stage detectors except for SSD and analyze their results on a single dataset, i.e. MS COCO dataset. Wu et al. (2020) delves into more recent advances in deep learning for object detection and provides information on the evolution of object detection from traditional methods to deep learning-based methods. They provide results mostly on two-stage detectors belonging to the R-CNN family. The accuracy is the main focus again while generalization, resource consumption, energy, robustness, and other metrics are omitted. The results are reported with resource-heavy backbones such as VGG-16 or ResNet-101 and do not qualify for a real-time detection survey.

All the survey papers provide an overview of detection over the years but there are shortcomings and gaps that have not been addressed. First, the newer design of detectors such as anchor-free and transformer-based detectors have not been considered. Second, metrics such as energy consumption, MAC count, and parameter count of the architectures are not compared to provide a more detailed analysis of the resources needed to use the networks. Third, other relevant capabilities such as robustness, reliability, and out-of-distribution generalization are omitted thus preventing acquiring a holistic analysis of all the detectors. Fourth, none of the surveys focus on real-time detection and hence cannot be used as a guide when trying to select (from existing networks) or design a new network for real-time applications.

We focus on real-time object detection and use a standardized (plug-&-play style) pipeline to evaluate a large number of feature extractors across anchor-, Keypoint-, and transformer-based detectors. We decouple the various factors that affect the detection results, such as image size, anchor size, deformable layers, and confidence thresholds, which helps in analyzing the networks in a more flexible manner. This facilitates a fair comparison framework to evaluate the detection models across a wide range of metrics and datasets, thus providing a holistic and unique insight into network design considerations. We, further, do an extensive robustness analysis against different corruptions and adversarial attacks. Robustness against adversarial attacks has been reported predominantly in the image recognition domain and not so much in object detection. To bridge this gap, we report the robustness performance of all the detection networks against varying attack strengths. We also perform a calibration study to evaluate the reliability of the predictions. Moreover, to analyze real-time performance, we benchmark all the networks on low-powered embedded devices, Nvidia TX2 and Xavier. Finally, we present two unique case studies on two diverse and equally important applications in Autonomous Driving and Healthcare domains.

## 3 Object Detection: Brief Overview

Object detection combines both classification and localization by providing class labels and bounding box coordinates of the object instances. Convolutional Neural Network (CNN) based object detectors are typically classified into two categories, viz. two-stage, and one-stage detection methods.

### 3.1 Two-stage Detection

The two-stage detectors consist of a separate Region Proposal Network (RPN) along with classification. The extracted features from the proposed Region of Interest (ROIs) in RPN are passed to both the classification head, to determine the class labels, and to the regression head to determine the bounding boxes (Girshick et al., 2014; Girshick, 2015; Ren et al., 2015; He et al., 2017; He et al., 2015). Region-based Convolutional Neural Network (RCNN) uses a selective search algorithm to find regions of pixels in the image that might represent objects and tthe shortlisted candidates are then passed through a CNN (Girshick et al., 2014). The extracted features from the CNN become inputs to a Support Vector Machine (SVM) that classifies the regions and to a regressor that predicts the bounding boxes. RCNN requires progressive multi-stage training and is quite slow. To overcome the shortcoming of RCNN, Fast-RCNN proposed certain modifications

Table 2: An overview of the details of the detection methods. The loss functions split up into classification (Cls) and localization (Loc) losses. Convolution layer types include vanilla 2D convolution (V), deformable convolution (DCN), bilinear upsampling (UP), depth-wise separable convolution (DWS), and transposed convolution (T).

| Head | Localization Type | | Multi-scale | | Neck Type | Convolution Layers | Loss Functions | | NMS Layer |
|------|-------------------|---|-------------|---|-----------|--------------------|----------------|---|-----------|
| | | | | | | | Cls | Loc | |
| Thundernet | two-stage | anchor-based | ✓ | 3 | CEM | V | CE | reg: Smooth L1 | Soft-NMS |
| Yolo | one-stage | anchor-based | ✓ | 2 | - | V | FL | reg: L2; conf: L2 | NMS |
| SSD | one-stage | anchor-based | ✓ | 6 | FPN | V | CE | reg: Smooth L1 | NMS |
| DETR | one-stage | self-attention | ✗ | - | - | - | CE | reg: L1 + GIoU | - |
| CenterNet | one-stage | keypoint-based | ✗ | - | - | V, DCN,T | FL | off: L1; emb: L1 | Maxpool |
| TTFNet | one-stage | keypoint-based | ✓ | 4 | - | V, DCN, UP | FL | reg: GIoU | Maxpool |
| FCOS | one-stage | keypoint-based | ✓ | 5 | FPN | V | FL | reg: IoU; cent: BCE | NMS |
| Nanodet | one-stage | keypoint-based | ✓ | 3 | PAN | V, DWS | GFL | reg: GIoU | NMS |

(Girshick, 2015). First, instead of using a CNN to extract features of the regions proposed by selective search, a CNN is used to directly extract features of the entire image. From the extracted features, a Region of Interest (ROI) pooling layer pools the features corresponding to the proposed image regions. Second, the SVM classifier and the regressor are each replaced with fully connected layers. Faster-RCNN proposed further improvements to get rid of the slow selective search algorithm for region proposals. The features extracted by the backbone CNN are sent to an additional CNN-based Region Proposal Network (RPN) that provides region proposals (Ren et al., 2015). However, despite high accuracy, the aforementioned two-stage detection methods are not suitable for real-time applications (Liu et al., 2018a; Zhao et al., 2018b; Zou et al., 2019). A lightweight two-stage detector named ThunderNet (Qin et al., 2019) was proposed with an efficient RPN coupled with a small backbone for real-time detection (Section 5.1).

### 3.2 One-stage Detection

One-stage detectors consist of a single end-to-end feedforward network performing classification and regression in a monolithic setting. These detectors do not have a separate stage for proposal generation and instead consider all positions on the image as potential proposals. Each of these is used to predict class probabilities, bounding box locations, and confidence scores. Confidence scores determines how sure the network is about its class prediction. The main two categories within one-stage detectors are the anchor-based and keypoint-based detectors.

Anchor-based detectors use predetermined anchor boxes (or priors) to aid predictions. Salient examples of this approach are You Only Look Once (YOLO; Redmon et al. (2016); Redmon & Farhadi (2017; 2018)) and Single Shot Detector (SSD; Liu et al. (2016)). YOLO works by visualizing the input image as a grid of cells, where each cell is responsible for predicting a bounding box if the center of a box falls within the cell. Each grid cell predicts multiple bounding boxes and outputs the location and the class label along with its confidence score. SSD was the first single-stage detector to match the accuracy of contemporary two-stage detectors while maintaining real-time speed. SSD predicts scores and box offsets for a fixed set of anchors of different scales at each location on several feature maps extracted from a Feature Pyramid Network (FPN). FPN facilitates multi resolution features (Lin et al., 2017a).

Anchor-based detectors have the disadvantage of dealing with hyperparameters such as number, aspect ratio, and size of the anchors which are highly dataset-dependent. This resulted in the introduction of a new paradigm of anchor-free (aka keypoint-based) object detectors. Keypoint-based methods treat objects as points instead of modeling them as bounding boxes. Keypoints, such as corners or center of objects, are estimated and the width and height are regressed from these points instead of predetermined anchors. Several keypoint-based networks were introduced, namely CornerNet, CenterNet, FCOS, NanoDet and TTFNet (Law & Deng, 2018; Zhou et al., 2019a; Tian et al., 2019; Lyu, 2020; Liu et al., 2020).

Although both anchor-based and keypoint-based detectors have achieved remarkable accuracy in generic object detection, it has largely been dominated by CNN-based architectures which lack global context.

Moreover, modern detectors typically perform regression and classification on a large set of proposals, anchors, or window centers. Thus, their performance suffers from complex post-processing tasks such as NMS (Non-Maximum Suppression to remove a near-identical set of proposals; more details in Section 4.1.3). Vision transformers have been introduced as an alternative architectural paradigm to CNNs. Transformer-based detectors, such as DETR (Carion et al., 2020), make use of self-attention modules which explicitly model all interactions between elements in a given sequence thus providing global context. The overall design of transformers also bypasses the hand-crafted processes such as NMS by making direct predictions on a given input.

Detailed information about each of these architectures is provided in Section 5 and Table 2.

## 4 Object Detection Formulation

Here, we first explain all the basic components of detection networks and then explain the main challenges in object detection.

### 4.1 Basic Components

The object detection problem can be formalized as: given an arbitrary image $\mathcal{I}_i$ and a predefined list of object classes, the object detection model not only classifies the type of object instances present in the image $\{c_1, c_2, ..., c_m\}$ but also returns the location of each object in the form of bounding boxes $\{b_1, b_2, ..., b_m\}$ where $b_i = \{(x_1, y_1), (x_2, y_2)\}$ is the top-left and bottom-right coordinates of the bounding box. Object detectors, both single-stage, and two-stage, typically consist of a feature extractor (hereafter, backbone) and detection head. A backbone is usually a CNN-based network that extracts the most prominent representations of a scene (from low to high-level features). Most backbones use pooling/convolution layers with strides to gradually reduce the size of feature maps and increase the receptive field of the network. The output feature maps are then passed to the detection head which performs classification and regression to determine the label and location of the object instances (Figure 2 shows an overview of a generic object detection).

### 4.1.1 Loss Functions

We present a brief overview of the loss functions used to train an object detector. Two objective functions are typically used to train CNN-based detector, i.e. the classification and the regression losses. The classification loss is usually defined by Cross-Entropy (CE) loss:

$$\mathcal{L}_{CE} = -\sum_{i=1}^{n} t_i \log(p_i) \tag{1}$$

where $t_i$ is the ground-truth label and $p_i$ is the softmax probability for the $i^{th}$ class. However, the CE loss does not account for imbalanced datasets where less frequent examples are much harder to learn compared to frequently occurring objects. Thus, Lin et al. (2017b) proposed Focal Loss (FL) which addresses the class imbalance by assigning more importance to the hard samples while down-weighting the loss contribution of easy samples,

$$\mathcal{L}_{FL} = -\alpha_i (1 - p_i)^\gamma \log(p_i) \tag{2}$$

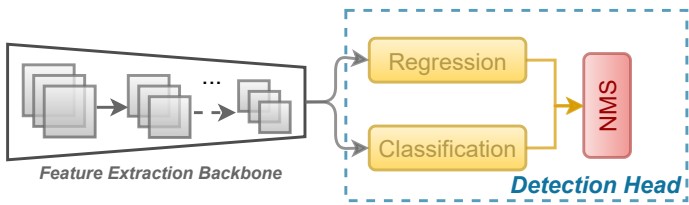

Figure 2: Schematic diagram of the main components of an object detector.

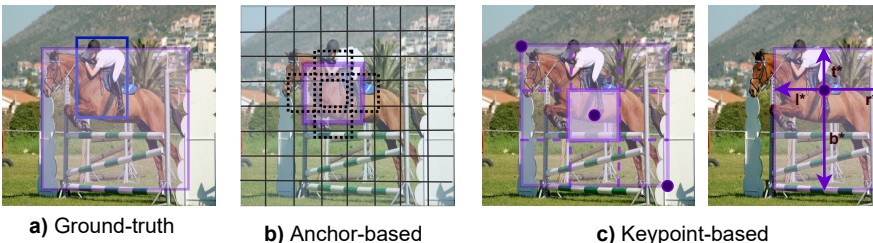

a) Ground-truth          b) Anchor-based          c) Keypoint-based

Figure 3: (**b**) Anchor-based techniques assign anchors to each grid in the image and make predictions as offsets to these. (**c**) The Keypoint-based techniques forego anchors and estimate keypoints (for instance, corners or center of object) onto heatmap and regress the width and height of the object from this keypoint.

where $\alpha_i$ is the weighting parameter, $\gamma \geq 0$ is the tunable modulating parameter.

The regression loss is usually an $L_1$ (Least Absolute Deviations) or $L_2$ (Least Square Error) loss on all the four bounding box coordinates between the ground-truth and the predicted bounding box.

### 4.1.2 Anchor Box vs. Keypoint

Anchor-based object detection techniques use the concept of anchor boxes (also interchangeably referred to as prior boxes in the literature). In this approach, an image is divided into grids in which each grid cell can be assigned to multiple predefined anchor boxes (Figure 3b). These boxes are defined to capture the scale and aspect ratio of specific object classes and are typically chosen based on object sizes in the training dataset. Intersection over Union (IoU) is calculated between the anchors and the ground-truth bounding boxes and the anchor that has the highest overlap is used to predict the location and class of that object. When overlap of an anchor with a ground-truth box is higher than a given threshold, it is considered as a positive anchor. The network, instead of directly predicting bounding boxes, predicts the offsets to the tiled anchor boxes and returns a unique set of predictions for each. The use of anchor boxes helps in detecting multiple objects, objects of different scales, and overlapping objects.

Anchor-based approaches, however, have two major drawbacks. First, a very large number of anchor boxes are required to ensure sufficient overlap with the ground-truth boxes and in practice, only a tiny fraction overlaps with the ground-truth. This creates a huge imbalance between positive and negative anchors, thus increasing training time. Second, the size, shape, and aspect ratio of anchor boxes are highly dataset-dependent and thus require fine-tuning for each dataset. However, such choices are made using ad-hoc heuristics (using the ground-truth boxes of the dataset) which become significantly more complicated with multi-scale architectures where each scale uses different features and its own set of anchors.

To mitigate the above-mentioned issues, keypoint-based object detection techniques are proposed that do not use anchors (Law & Deng, 2018; Zhou et al., 2019a; Tian et al., 2019). The detection problem is reformulated as a per-pixel prediction, similar to segmentation. The features of a CNN are used to create heatmaps, where the intensity peaks represent keypoints such as corners or centers of relevant objects. Along with these, there are additional branches that predict the dimensions of the boxes (width and height). The heatmap predictions along with embeddings are used to estimate the correct location and size of the predicted boxes. For center keypoints, distances from centers to the four sides of the object bounding box are predicted for detection (Figure 3c).

### 4.1.3 Non-Maximum Suppression (NMS)

Object detectors produce far too many proposals, many of which are redundant. To remove the dense set of duplicate predictions, detectors commonly use a post-processing step called NMS. The NMS module first sorts the predicted proposals for each instance by their confidence score and selects the proposal with the highest confidence. Subsequently, a Jaccard overlap is performed with the near-identical set of proposals for

each instance,

$$IoU(b_m, b_i) = \frac{b_m \cap b_i}{b_m \cup b_i} \tag{3}$$

where $b_m$ is the proposal with the highest confidence and $b_i$ represents each of the near-identical proposals for the same instance. The duplicates are removed if this value is greater than the set NMS threshold (typically 0.5).

However, one of the associated issues with NMS is that valid proposals are rejected when the proposals (for different instances) are close to each other or overlapping in some cases. This is especially true for crowded scenes. Therefore, Soft-NMS was proposed to improve the NMS constraint. In Soft-NMS, the scores of detection proposals (for other instances) which has slightly less overlap with $b_m$ are decayed while ensuring that the proposals that have higher overlap with $b_m$ are decayed more, such that the duplicates can be removed. This is done by calculating the product of the confidence score (of a proposal $b_i$), and the negative of IoU with $b_m$ as shown below:

$$s_i = \begin{cases} s_i, & \text{if } IoU(b_m, b_i) < threshold \\ s_i(1 - IoU(b_m, b_i)), & \text{if } IoU(b_m, b_i) \geq threshold \end{cases} \tag{4}$$

Keypoint-based detectors do not use this IoU-based NMS as they process points on heatmaps instead of overlapping boxes. The NMS in these networks is a simple peak-based maxpool operation that is computationally less expensive.

### 4.2 Challenges of Object Detection

Object detection as a computer vision problem is inherently challenging as an ideal detector has to deliver both high accuracy and high performance at reasonable energy and computational expense. Later in Section 8, we discuss the pros and cons of several backbone and detection head combinations to demonstrate the trade-off between accuracy and speed.

Detection accuracy and inference speed are also dependent on both image sizes and object sizes. While higher image resolutions yield better accuracy by extracting more information from the scene, it also lowers inference speed. Therefore, choosing an image size that provides the right balance between accuracy and speed is of paramount importance. Additionally, object sizes play a major role in detection accuracy. While detectors can achieve high accuracy on medium to large objects, almost all detectors struggle to detect smaller objects in the scene (Liu et al., 2021). In Sections 9.1 and 9.2, we study the effects of object sizes and image sizes on detection accuracy and speed.

To deliver high accuracy, the detector needs to be robust and localize and classify real-world objects with significant intra-class variation (for instance, variations in size, shape, and type of objects), pose, and non-rigid deformations. For using anchor-based detectors, the optimization of anchors is a challenge as they are dataset dependent. In Section 9.3, we demonstrate how varying anchor sizes affect detection accuracy.

Another major challenge is maintaining consistent performance in varying weather (rain, snow, blizzard) and lighting conditions. For applications such as autonomous driving, the detector also has to account for cluttered backgrounds, crowded scenes, and camera effects. We provide a detailed study on the robustness of the detectors in Section 11.

Finally, deep neural networks tend to rely on shortcut learning cues, hence overfitting to the training data distribution and not generalizing to out-of-distribution (OOD) data. As generalization to the unseen data is the most important concern, we provide a detailed analysis on both in-distribution as well as OOD data in Section 13.

## 5 Object Detection Heads

Since the scope of our study is real-time object detection, we focus on one **two-stage** detector: ThunderNet (Qin et al., 2019), two **anchor-based** detectors: SSD (Liu et al., 2016), YOLO (Redmon & Farhadi, 2017),

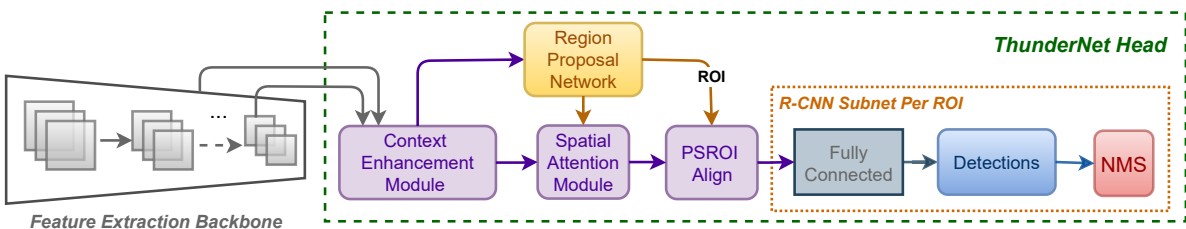

Figure 4: Schematic diagram of two-stage detector, ThunderNet.

four (anchor-free) **keypoint-based** detectors: CenterNet (Zhou et al., 2019a), FCOS (Tian et al., 2019), NanoDet (Lyu, 2020) and TTFNet (Liu et al., 2020), and one **transformer-based** detector: DETR (Carion et al., 2020).

## 5.1  ThunderNet

ThunderNet revisits the two-stage detector architecture and improves upon Lighthead-RCNN (Li et al., 2017) and uses a variant of ShuffleNet-v2 (Ma et al., 2018) as backbone. The detection head increases the number of channels in the early stages of the network to encode low-level features and provide a boost in accuracy. ThunderNet uses two new modules: Context Enhancement Module (CEM) and Spatial Attention Module (SAM). CEM aggregates features from three different scales enlarging the receptive field by utilizing local and global features. SAM refines the features by strengthening foreground features while suppressing the background features (Figure 4). The output of the SAM module is given as:

$$F_{SAM} = F_{CEM} \cdot \sigma(\mathcal{T}(F_{RPN})) \tag{5}$$

where $F_{SAM}$, $F_{CEM}$, and $F_{RPN}$ represent the output features of SAM, CEM, and RPN modules, respectively. $\sigma(.)$ is the sigmoid function and $\mathcal{T}(.)$ denotes the dimension transformation function to match the number of output channels from $F_{CEM}$ and $F_{RPN}$.

$$\mathcal{L}_{rpn} = \frac{1}{N_b} \sum_i \mathcal{L}_{cls}(p_i, t_i) + \lambda \frac{1}{N_a} \sum_i t_i \mathcal{L}_{reg}(b_i, b_g) \tag{6}$$

where $\mathcal{L}_{cls}$ is the log loss over two classes (object or not). $b_i$ and $b_g$ represent the bounding box regression targets for anchor $i$ and the ground-truth, respectively. Anchors with overlap with any ground-truth box higher than a given threshold are considered positive ($t_i = 1$) and the rest of the anchors are considered as negative ($t_i = 0$). Thus, the multiplicative term $t_i$ ensures that the regression loss is activated only for positive anchors. $N_a$ and $N_b$ represent the number of anchor locations and mini-batch size, and $\lambda$ is the balancing weight. Similar to Fast R-CNN, ROI pooling is performed and these regions are sent through two branches for classification and regression with objective functions as follow:

$$\mathcal{L} = \mathcal{L}_{cls}(p, u) + \lambda[u \geq 1]\mathcal{L}_{reg}(b_u, b_g) \tag{7}$$

where $\mathcal{L}_{cls}$ is the log loss for true class u and $\lambda$ is the balancing weight. $\mathcal{L}_{reg}$ computes the regression loss over the ground-truth bounding box and predicted box for class $u$. $[u \geq 1]$ is the Inverson indicator function that evaluates to 1 when $u \geq 1$ ($u = 0$ is the background class).

## 5.2  You Only Look Once (YOLO)

YOLO (Redmon et al., 2016) is a single-stage object detection network that was targeted for real-time processing. YOLO divides image into grid cells and each cell predicts one object defined by bounding boxes and scores. An object is said to belong to a particular grid cell if its center lies in it. YOLO is fast and simple but suffers from low recall. Redmon & Farhadi (2017) proposed YOLO v2 to improve accuracy and speed of YOLO. Instead of making arbitrary guesses for bounding boxes, YOLO v2 uses anchors of different sizes and aspect ratios in each grid to cover different positions and scales of the entire image. Anchors can be

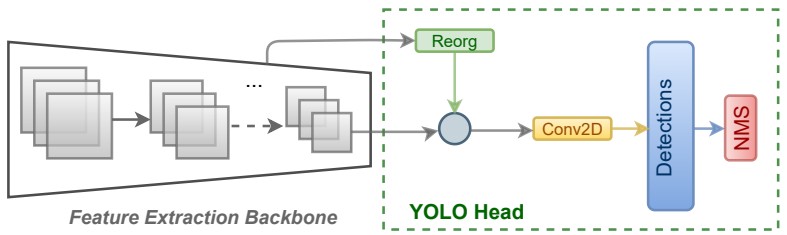

Figure 5: Schematic diagram of single-stage anchor-based detectors, YOLO.

made more accurate by calculating anchor sizes using an IoU-based k-Means clustering over the particular dataset. The network predictions are the offsets of each of the anchor boxes. YOLO v2 makes bounding box predictions on a single feature map obtained by concatenating two feature map scales (Figure 5). Multiple other versions of YOLOs have been introduce and all of them are built upon the fundamental concept of YOLO v2 but with many tips and tricks to achieve higher performance. As we are trying to evaluate on a fair and simple framework, in this study, we consider YOLO v2 version only, as it is simple, fast, and has the minimal bag of tricks.

The loss function composes of the classification loss, the localization loss, and the confidence loss (which measures the objectness of the box):

$$\mathcal{L} = \mathcal{L}_{cls} + \lambda\mathcal{L}_{reg} + \lambda'\mathcal{L}_{conf} \tag{8}$$

where $\mathcal{L}_{cls}$ is Focal Loss, $\mathcal{L}_{reg}$ and $\mathcal{L}_{conf}$ are both $L_2$ losses. $\mathcal{L}_{conf}$ is the confidence loss that measures the objectness of a box ( Ex. if an object is not present in the bounding box, then its confidence of objectness will be reduced. $\lambda$ and $\lambda'$ are the balancing weights.

### 5.3 Single Shot Multibox Detector (SSD)

SSD (Liu et al., 2016) has a feedforward CNN which produces bounding boxes, confidence scores and classification labels for multiple object instances in a scene. SSD uses multiple feature maps from progressively reducing resolutions emulating input images at different sizes, at the same time sharing computation across scales. While the feature maps from shallow layers are used to learn the coarse features from smaller objects, the features from deeper layers are used to localize larger objects in the scene.

The detection head employs separate pre-defined anchor boxes for each feature map scale and combines the predictions of all default anchors at different scales and aspect ratios (Figure 6). The scale and size of the anchors for each feature map $k$ is defined as:

$$s_k = s_{min} + \frac{s_{max} - s_{min}}{m - 1}(k - 1) \tag{9}$$

where $k \in [1, m]$ and the default values for $s_{min}$ and $s_{max}$ are given as 0.2 and 0.9, respectively. $m = 6$ feature maps are used in SSD.

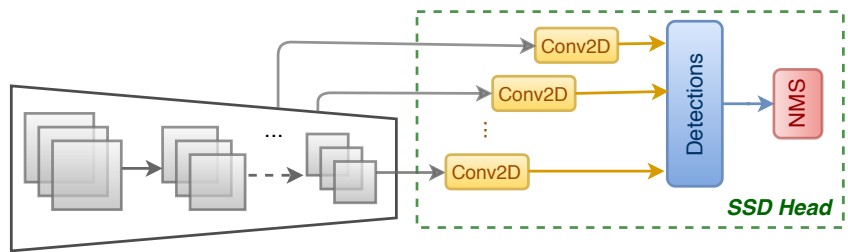

Figure 6: Schematic diagram of single-stage anchor-based detector, SSD.

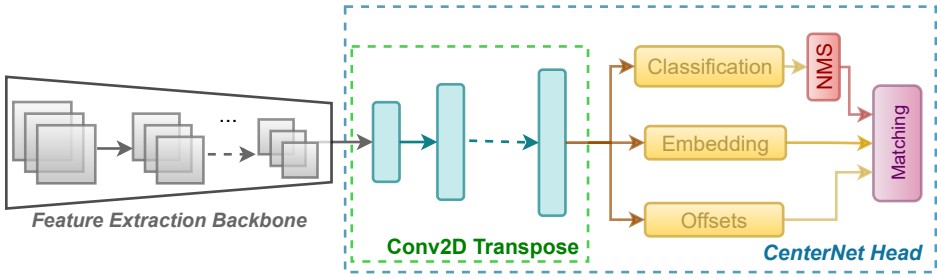

Figure 7: Schematic diagram of single-stage keypoint-based detector, CenterNet.

SSD produces a diverse set of predictions covering object instances of various shapes and sizes. SSD uses a matching strategy to determine which anchors correspond to the ground-truth and the anchor with the highest overlap is used to predict that object's location and class. The objective function is derived from multibox objective (He et al., 2015) and extended for multiple categories. The overall objective function is

$$\mathcal{L} = \frac{1}{N_{pos}}\mathcal{L}_{cls} + \lambda\mathcal{L}_{reg} \tag{10}$$

where $\mathcal{L}_{cls}$ is the cross-entropy loss, $\mathcal{L}_{reg}$ is the sum of Smooth $L_1$ loss across all bounding box properties for matched positive boxes. $N_{pos}$ is the number of positive samples and $\lambda$ is the balancing weight.

### 5.4 CenterNet

Anchor-based detectors have to deal with hyperparameters such as number, aspect ratio, and size of anchors that are also highly dataset dependent. CornerNet was proposed as the first alternative to the anchor-based approach that reduced the object detection problem to keypoint estimation problem (Law & Deng, 2018).

Amongst the multiple methods proposed in (Law et al., 2019; Zhou et al., 2019b;a), we use CenterNet (Zhou et al., 2019a), since it not only achieves higher accuracy than CornerNet, but also simplifies keypoint estimation. The detection algorithm augments the backbone with three transposed convolutional layers to incorporate high resolution outputs. The first branch outputs a heatmap to estimate the keypoints or the center points of the objects and the number of heatmaps is equal to the number of classes. Ground-truth heatmaps are created by using Gaussian kernels on the centers of the ground-truth boxes. The peaks are used to estimate the center of an instance and also determine the instance category. There are two more branches generating heatmaps: the embedding branch regresses the dimensions of the boxes, i.e. width and height, and the offset branch accounts for the discretization error caused by mapping the center coordinates to the original input dimension (see Figure 7). The overall objective function is given as:

$$\mathcal{L} = \mathcal{L}_{cls} + \lambda\mathcal{L}_{off} + \lambda'\mathcal{L}_{emb} \tag{11}$$

where the $\mathcal{L}_{cls}$ is a penalty reduced pixel-wise logistic regression with Focal Loss (Lin et al., 2017b), $\mathcal{L}_{off}$ is an $L_1$ loss to minimize the discretization on the center coordinates and finally $\mathcal{L}_{emb}$ is also an $L_1$ loss to minimize errors in computing the width and height of the predicted boxes. $\lambda$ and $\lambda'$ are balancing weights.

### 5.5 Fully Convolution One-Stage Object Detection (FCOS)

FCOS, a fully convolutional anchor-free detector, reformulates object detection as a per-pixel prediction problem similar to semantic segmentation (Tian et al., 2019). The detector uses multi-level prediction with FPN (Lin et al., 2017a) to improve recall and resolve the ambiguity from overlapping bounding boxes. Five feature maps are obtained at different scales and pixel-by-pixel regression is performed on each of these layers. This increases recall but gives rise to low-quality predictions at locations far away from the center of the object. To avoid this, an additional branch is added in parallel, to predict the centerness of a location (Figure 8). The overall loss function is given as:

$$\mathcal{L} = \frac{1}{N_{pos}}\mathcal{L}_{cls} + \frac{\lambda}{N_{pos}}\mathcal{L}_{reg} + \mathcal{L}_{cent} \tag{12}$$

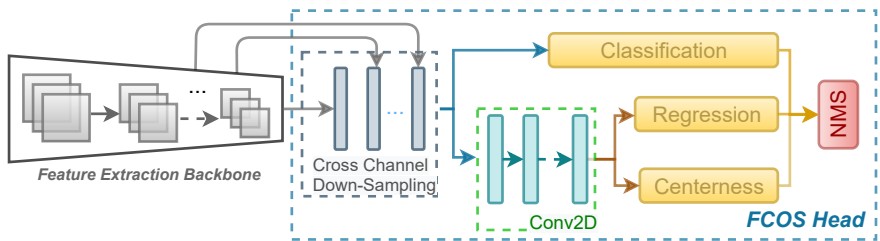

Figure 8: Schematic diagram of fully convolutional single-stage keypoint-based detector, FCOS.

where $\mathcal{L}_{cls}$ is Focal Loss, $\mathcal{L}_{reg}$ is IoU regression loss, and $\mathcal{L}_{cent}$ is the centerness loss which uses a Binary Cross Entropy (BCE) loss. $N_{pos}$ is the number of positive samples and $\lambda$ is the balancing weight.

The IoU regression is based on UnitBox (Yu et al., 2016) and is a form of cross-entropy loss with input as IoU. Instead of the $L_2$ loss that optimizes the coordinates independently, the IoU loss treats it as one unit. The final score is weighted by the centerness-score. Hence, this branch down-weighs the scores of bounding boxes predicted farther away from the object center, which helps the final NMS in filtering out low-quality predictions.

### 5.6 NanoDet

Inspired by FCOS, NanoDet was introduced as a lightweight anchor-free detector (Lyu, 2020). The proposed detector uses the Adaptive Training Sample Selection (ATSS) module (Zhang et al., 2020) which automatically selects positive and negative training samples based on object characteristics. The detector uses a Generalized Focal Loss (GFL) (Li et al., 2020) for classification and regression. GFL aims to extend Focal Loss from discrete to the continuous domain for better optimization. This is a combination of Quality Focal Loss (QFL) and Distributed Focal Loss (DFL). QFL combines classification and IoU quality to provide one score for both and DFL views the prediction boxes as a continuous distribution and optimizes it. Generalized IoU loss (GIoU) is useful for non-overlapping cases, as it increases the predicted box's size to overlap with the target box by moving slowly towards the target box. The overall loss function used for training NanoDet is given as:

$$\mathcal{L} = \frac{1}{N_{pos}} \sum_z \mathcal{L}_{QFL} + \frac{1}{N_{pos}} \sum_z \mathbb{1}_{\{c_z^* > 0\}}(\lambda \mathcal{L}_{GIoU} + \lambda' \mathcal{L}_{DFL}) \tag{13}$$

where $\mathcal{L}_{QFL}$ and $\mathcal{L}_{DFL}$ are the Quality and Distributed Focal losses and the $L_{GIoU}$ is the Generalized IoU loss. $N_{pos}$ is the number of positive samples and $\lambda$ and $\lambda'$ are balancing weights. $z$ denotes all locations on the pyramid feature maps.

While FCOS uses five feature maps which are passed to a multi-level FPN, NanoDet uses three feature maps which are passed to three individual Path Aggregation Network (PAN) (Liu et al., 2018b) blocks. PAN is similar to FPN but enhances the lower-level features by adding a bottom-up path augmentation. The outputs from the PAN blocks are connected to individual detection heads which compute the classification labels and bounding boxes for a specific feature map. NanoDet also removes the centerness branch which

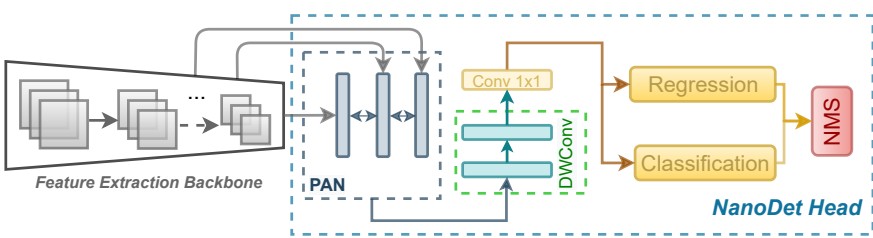

Figure 9: Schematic diagram of fully convolutional single-stage keypoint-based detector, NanoDet.

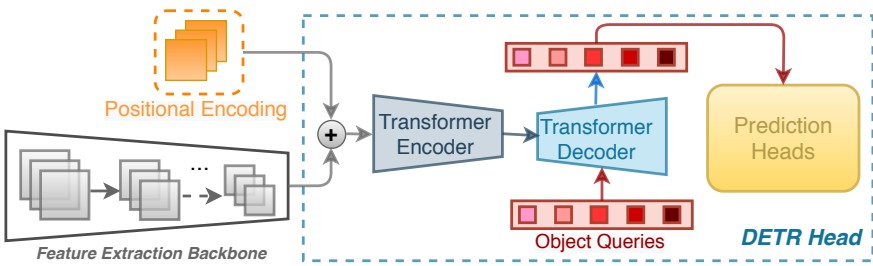

Figure 10: Schematic diagram of transformer-based detector, DETR.

exists in FCOS, thus making it a faster variant. The outputs from the three heads are finally passed to the NMS to achieve the final boxes and classification labels for the input image (see Figure 9).

## 5.7   DEtection TRansformer (DETR)

Transformers are a new design paradigm in computer vision which rely on attention mechanism and were introduced for the first time in object detection by DETR (Carion et al., 2020). DETR casts the object detection task as a direct set prediction problem which eliminates duplicate bounding box predictions. Transformers capture the pairwise relations between the objects based on the entire image context by using the self-attention module, thus avoiding repetitive predictions. This is a benefit over the conventional object detectors which use post-processing steps such as NMS to eliminate the duplicate predictions, hence creating a computational bottleneck.

DETR consists of an encoder-decoder transformer and a FeedForward Network (FFN) that makes the final prediction (Figure 10). The encoder consists of a Multi-Head Self-Attention (MHSA) module (Vaswani et al., 2017) and a FFN. These blocks are permutation invariant and hence, fixed positional encoding is added to the input of every attention layer. The decoder uses the encoder features and transforms the object queries to an output embedding using multiple MHSA modules. The $N$ output embeddings are used by two different FFN layers, one for predicting class labels and the other one for predicting box coordinates. DETR finds the best predicted box for every given ground-truth using unique bipartite matching. The one-to-one mapping of each of the $N$ queries to each of the $N$ ground-truths is computed efficiently using the Hungarian optimization algorithm. After obtaining all matched pairs for the set, a standard cross-entropy loss for classification and a linear combination of $L_1$ and GIoU losses for regression are used. Auxiliary losses are added after each decoder layer to help the model output the right number of objects in each class. Given $\lambda$ and $\lambda'$ are the balancing weights, the total loss is as follow:

$$\mathcal{L} = \lambda \mathcal{L}_{cls} + \lambda' \mathcal{L}_{reg} \tag{14}$$

## 5.8   Training-Time-Friendly Network (TTFNet)

Inspired by CenterNet (Zhou et al., 2019a), TTFNet uses the same strategy wherein detection is treated as a two-part problem of center localization and bounding box size regression (Liu et al., 2020). For center localization, TTFNet adopts the Gaussian kernel to produce a heatmap with higher activation near the object center similar to CenterNet, but additionally, it also considers the aspect ratio of the bounding boxes. For size regression, instead of choosing just the center pixel as a training sample, TTFNet proposes to treat all the pixels in the Gaussian area as training samples. Also, these samples are weighted by the weight calculated by the target size and Gaussian probability, thus utilizing more information. The motivation is that encoding more training samples is similar to increasing the batch size which helps in expanding the learning rate and speeding up the training process.

TTFNet modifies the Gaussian kernel by constructing a sub-region around the center of an object and extracting training samples only from this region (Figure 11). Gaussian probability is used as the weight to emphasize samples close to the center of the target hence alleviating the overlapping ambiguity. Due to the large variance in object sizes, larger objects produce significantly more samples than smaller ones and hence

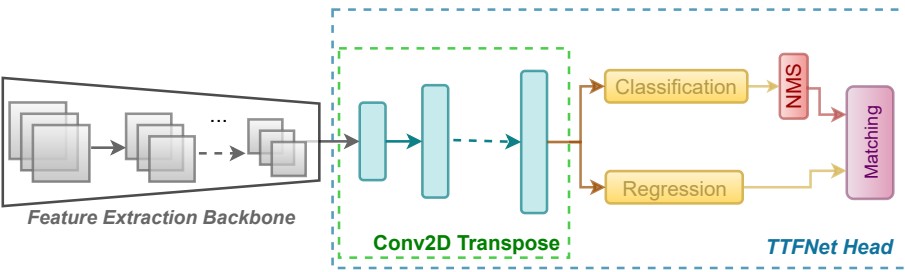

Figure 11: Schematic diagram of single-stage keypoint-based detector, TTFNet.

the contribution of smaller objects is negligible which can hamper detection accuracy. Thus, a loss balancing strategy is introduced which makes good use of more annotation information in large objects while retaining the information for smaller objects.

$$\mathcal{L}_{reg} = \frac{1}{N} \sum_{(i,j) \in A_m} GIoU(\hat{b}_{ij}, b_m) \times W_{ij} \tag{15}$$

Given ground-truth box $b_m$, Gaussian kernel is adopted and each pixel within a sub-area $A_m$ is treated as a regression sample. $\hat{b}_{ij}$ is the predicted box, $W_{ij}$ is the balancing weight, and $N$ is the number of regression samples. Thus, the Overall loss is as below:

$$\mathcal{L} = \lambda \mathcal{L}_{cls} + \lambda' \mathcal{L}_{reg} \tag{16}$$

where $\lambda = 1.0$ and $\lambda' = 5.0$ are classification and regression balancing weights and $\mathcal{L}_{cls}$ is a modified version of the Focal Loss presented in Kong et al. (2019).

## 6 Backbones

In this study, we choose nine feature extractors as backbones based on factors such as speed, energy, and memory efficiency specifically catered for real-time applications. In the following, we present the backbones in chronological order.

**ResNet:** He et al. (2016) reformulated the network layers as learning residual functions with residual skip connections. Networks with skip connections are easier to optimize and can gain considerable accuracy with increased depth. *ResNet-18*, a lightweight variant of deep residual networks consisting of four residual blocks each with two convolutions followed by batch normalization layers, is considered in this study.

**DarkNet:** Redmon & Farhadi (2017) proposed a computationally lightweight feature extractor DarkNet as part of their proposal for a real-time object detection algorithm, YOLO. Darknet improves over VGG-16 by reducing the number of parameters. For the purpose of real-time detection, only *DarkNet-19* is considered in this study.

**Xception:** Chollet (2017) proposed *Xception* as an improvement to Inception-V3 (Szegedy et al., 2016b) which is entirely based on depth-wise separable convolutions (DWS; Kaiser et al. (2017)). The proposed architecture is a linear stack of 36 depth-wise separable convolution layers, structured into 14 modules all of which have residual connections, except the first and last.

**MobileNet:** Sandler et al. (2018) designed *MobileNet-v2* as a lightweight backbone specifically for real-time object detection on embedded devices. The architecture uses inverted residual blocks with linear bottlenecks and depth-wise separable convolutions. It is termed as inverted because skip connections exist between narrow parts of the network which results in a lesser number of parameters. Additionally, the network contains skip connections for feature re-usability between the input and output bottlenecks.

Table 3: A detailed list of the datasets containing the total number of images and annotations with the breakdown of the training and test sets, the number of classes (#cls), image resolution, and highlight of the type of data. U and R stand for urban and residential areas, whereas D and N stand for day and night cases.

| Dataset | Year | Type | #cls | Resolution | Total Img (annotations) | Training Img (annotations) | Test Img (annotations) |
|---|---|---|---|---|---|---|---|
| VOC | '07, '12 | Generic | 20 | Various | 21,503 (52,090) | 16,551 (40,058) | 4952 (12,032) |
| COCO | '17 | Generic | 80 | Various | 123,287 (896,782) | 118,287 (860,001) | 5000 (36,781) |
| BDD | '18 | AD: U/R/D/N | 10 | 1280 x 720 | 80,000 (1,472,397) | 70,000 (1,286,871) | 10000 (185,526) |
| Cityscapes | '16 | AD: U/D | 10 | 2048x1024 | 5000 (99,523) | 4500 (83,574) | 500 (15,949) |
| Kvasir-SEG | '20 | Medical | 1 | various[1] | 1000 | 800 | 200 |

**ShuffleNet-v2:** Ma et al. (2018) designed *ShuffleNet-v2* to optimize inference latency by reducing the memory access cost. The building blocks of this architecture consist of a channel split operation that divides the input into two, each of which is fed to a residual block. A channel shuffle operation is introduced to enable information transfer between the two splits to improve accuracy. The high efficiency in each building block enables using more feature channels and larger network capacity.

**VoVNet:** Lee et al. (2019) proposed VoVNet as an energy efficient backbone for real-time detection. It is built using One-Shot Aggregation (OSA) modules, that concatenate all intermediate features only once in the last feature map. The OSA block has convolution layers with the same input/output channel for minimizing the MAC count, which in turn improves the GPU computation efficiency. *VoVNet-39*, the faster and the more energy efficient variant, is used in this study.

**EfficientNet:** Tan & Le (2019) designed EfficientNet, a feature extractor, using an automated multi-objective architecture search which is optimized for accuracy and MAC count. The proposed architecture achieves high accuracy by re-adjusting and balancing network depth, width, and resolution. The building blocks of this architecture uses Mobile Inverted Bottleneck Convolutions (MBConv) and also includes Squeeze and Excitation (SE) modules (Hu et al., 2018). Amongst the several versions proposed, we use *EfficientNet-B0* which is the most lightweight version of this architecture.

**HarDNet:** Chao et al. (2019) proposed Harmonic Densely Connected Networks (HarDNet) to achieve high efficiency in terms of both MAC count and memory traffic. HarDNet stands out among all the other backbones in terms of reduced DRAM (Dynamic Random Access Memory). The sparsification scheme proposes a connection pattern between the layers such that it resembles an overlapping of power-of-two harmonic waves (hence the name). The proposed connection pattern forms a group of layers called a Harmonic Dense Block (HDB) and instead of considering all layers, gradients in a HDB with depth $L$ pass through only 'log $L$' layers. The output of a HDB is the concatenation of layer $L$ and all its preceding odd number layers and the output of even layers is discarded once the HDB is done. Also, stride-8 is used instead of stride-16 (that is adopted in many CNN networks) to enhance local feature extraction. Along with the reduction of the traffic of feature maps, it also provides other advantages such as low latency time, higher accuracy, and higher speed. Amongst the several versions proposed, *HarDNet-68* is used for this study.

**DeiT:** Touvron et al. (2021) modified the vision transformers to be used as a feature extractor for dense prediction tasks. The proposed architecture, Data-efficient image Transformer (DeiT), consists of repeated blocks of self-attention, feedforward layers, and an additional distillation module. To extract meaningful image representations, the learned embeddings from the final Transformer block are sent to an extra block to get features at different scales before sending it to the detection head. The smallest version of this architecture, i.e. *DeiT-T* (where T stands for tiny) is used for this study.

---

[1]ranging from 332×487 to 1920×1072

## 7 Empirical Evaluation

We briefly introduce the datasets, the evaluation metrics, and the experimental setup. The overall summary of all datasets is shown in Table 3. The experimental setup and hyperparameter details for all the detectors are outlined in Appendix, Table 12.

### 7.1 Datasets

PASCAL VOC, henceforth termed as **VOC** (Everingham et al., 2010), consists of 20 object categories divided into two datasets viz. VOC 2007 and 2012 with a combined total of $21,493$ images containing $52,090$ annotations. We combine the training and validation sets of both VOC '07 and '12 for training. The final testing is conducted over the VOC '07 test set consisting of 4952 images.

**COCO** (Lin et al., 2014) is a more challenging dataset and consists of 80 object categories. Following Zhou et al. (2019a), we use the 2017 split of the dataset which contains $118,287$ images ($860,001$ labeled instances) for training. For the final testing, we use 5000 images with $36,781$ instances.

**BDD** (Yu et al., 2018) is one of the largest and most challenging datasets on autonomous driving. It contains a variety of driving scenarios including urban, highways, and rural areas as well as a variety of weather and day/night driving conditions representing real-life driving challenges. The training set contains $69,863$ images with $\approx 1.28$ million labeled instances and the test set contains $10,000$ images with $185,526$ labeled instances across 10 object categories.

**Cityscapes** (Cordts et al., 2016) is a well-documented dataset for complete scene understanding in challenging urban scenarios. We extracted the bounding boxes from the instance segmentation and the labeled annotations were then grouped into 10 super-categories to match the categories of BDD. For OOD evaluation, we used the test set of 500 images with 15949 bounding boxes.

**Corrupted COCO.** To test the robustness of the models, we create a dataset that emulates different external influences that are found in real-world scenarios, by adding corruptions (Michaelis et al., 2019) to the original COCO dataset. There are 15 different corruptions and we categorize these into four groups: Noise, Blur, Weather, and Digital effects. Noise consists of Gaussian, Impulse, and Shot noise. The Blur group consists of Defocus, Glass, Motion, and Zoom blur effects. We use Brightness, Fog, Frost, and Snow to mimic different Weather conditions. Finally, we account for Digital effects by adding changes in Contrast, Elastic Transformation, JPEG compression, and Pixelation. These 15 corruptions are applied at 5 different levels of severity. The severity level ranges from 1 (less severe corruption) to 5 (most severe corruption).

**Kvasir-SEG** (Jha et al., 2020) is a bio-medical dataset for localizing polyps in gastrointestinal region. The dataset consists of 1000 images with the segmentation masks of polyps present in each image. The dataset also has bounding boxes obtained from the segmentation masks. Here, the dataset is divided into 800 images for training and 200 images for testing.

### 7.2 Evaluation Metrics

Object detectors make predictions in terms of a bounding box and a class label. Here, we first measure the overlap between the predicted bounding box and the ground-truth by calculating the IoU (given in Eq. 3). Based on a IoU threshold, the predicted box is categorized as True Positive (TP), False Positive (FP), or False Negative (FN). Next, we compute the precision and recall,

$$
\begin{aligned}
precision &= \frac{TP}{TP + FP}, \\
recall &= \frac{TP}{TP + FN}.
\end{aligned}
\tag{17}
$$

Precision measures how accurate the predictions are while recall shows how well the model finds all the positives. High precision but low recall implies more FN while the reverse implies more FP. A precision-recall (PR) curve shows the trade-off between the precision and recall values for different thresholds. It is

downward sloping because as the threshold is decreased, more predictions are made (high recall) and the less precise they are (low precision). We compute the average of precisions (AP) across all the recall values (between 0 and 1) at various IoU thresholds which can be interpreted as the area under the PR curve. Finally, the mAP (mean Average Precision) is computed by averaging the AP over all the classes. The PASCAL VOC challenge (Everingham et al., 2010) evaluates the mAP at 0.5 IoU threshold (@IoU:0.5) whereas the COCO challenge (Lin et al., 2014) sets ten different thresholds @IoU:[0.5-0.95] using 0.05 step size. In some applications like healthcare, recall measure holds more value as having more FN is more harmful than FP. Average Recall is measured by averaging recall over all IoUs, and the average of these is reported as the mean Average Recall (mAR). We also report the F1 score metric, which measures the balance between precision and recall. The F1 score is computed as follows:

$$F1 = 2 \times \left( \frac{precision \times recall}{precision + recall} \right) \tag{18}$$

We compute the MAC (multiply-accumulate operations) count for the convolution, batch norm, and fully connected layers for the detector and also report the number of learnable parameters (in million scales). We also report the inference speed in frames per second (FPS) for the combination of each backbone and head. The inference speed is computed for 500 images and averaged to remove any bias.

Finally, given the recent trend towards energy-efficient AI (Schwartz et al., 2019), we compute the inference energy consumption of a model on the whole test dataset. We use the NVIDIA Management Library (NVDIA, 2019) to calculate the approximation power consumption by the GPU during the inference. The inference energy consumption on a dataset is reported in kilo Joules (KJ) and we do not include the power consumption of other components.

### 7.3 Experimental Setup

All the networks are re-implemented in our repository. The original results from the corresponding papers are reproduced prior to performing other experiments. Our complete framework is in PyTorch 1.7 (Paszke et al., 2019) and includes all the backbones and detection heads to execute all the training and evaluations. For a fair comparison between different object detection networks, we use a consistent and uniform scheme for both training and evaluation.

Unless otherwise stated, all our experiments use the following training scheme. It is important to note that some of the detection heads (such as YOLO and DETR) use multi-scale training in their original implementations, but for a uniform training scheme, we use single-scale training with an image size of 512. All images are first normalized by per-channel mean subtraction using ImageNet mean values (Russakovsky et al., 2015) dataset. Default PyTorch weight initialization, with a fixed seed value, is used for detection heads, and pre-trained ImageNet weights are used for the backbones. For data augmentation, we use to expand, random horizontal flip, random crop, and random photometric distortions which include random contrast within the range of [0.5, 1.5], saturation [0.5, 1.5], and hue [-18, +18].

We use a batch size of 32 and train the models using Stochastic Gradient Descent (SGD) optimizer (Bottou, 2010) with 0.9 momentum with a learning rate decay factor of 0.1. The learning rate scheduler is chosen to ensure the convergence of all the models. The only exception to this rule is DETR where we follow the authors and use AdamW (Loshchilov & Hutter, 2017) optimizer. The NMS threshold is set to 0.45 and the confidence threshold to 0.01 wherever applicable. For all the experiments, we evaluate the models on an NVIDIA RTX 2080Ti GPU unless otherwise stated. The Pytorch models are converted to their optimized high-performance inference engines, using NVIDIA TensorRT (version 8.0), to facilitate deployment on embedded hardware. The TRT conversion optimizes the network by fusing multiple layers in the network (including convolution and batch normalization layers) to enable parallel processing. The inference energy consumption is calculated using the NVIDIA NVML API (Corporation, 2020) on a single clean machine with minimal running applications.

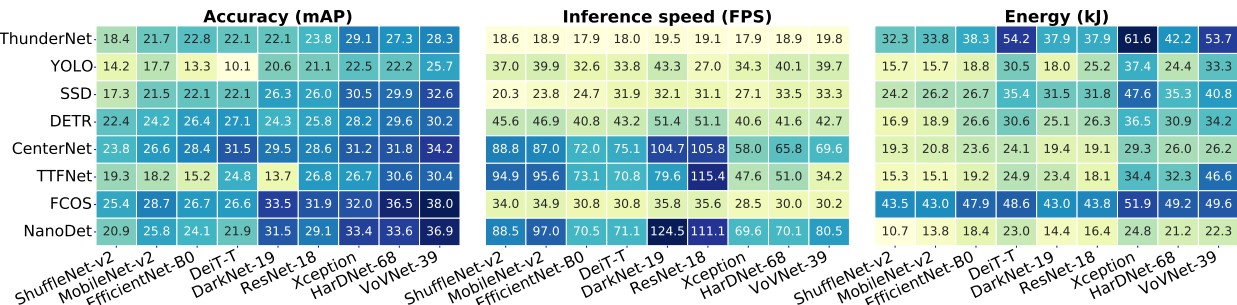

Figure 12: Summary result of all detection models with various backbones on the COCO dataset reported on three evaluation metrics - accuracy, speed, and energy. The darker shade in accuracy and inference speed and lighter shade in energy represents an ideal case.

## 8 Results

We provide a broad overview of results for all networks in Figure 12 to demonstrate the overall trend for three metrics: inference accuracy, speed, and energy consumption on the COCO dataset. The backbones and detection heads are ordered taking "accuracy" as the pivotal metric and sorting the results from low to high [1]. For easier performance analysis, we categorize backbones and heads into three spectra, namely low, middle and high, where low and high are the least and most accurate backbone and head combinations and the middle spectrum contains networks that have average performance.

**Accuracy**: Figure 12 demonstrates that the three backbones, i.e. *VoVNet-39*, *HarDNet-68* and *Xception* consistently achieve high accuracy across all heads and belong to the high spectrum. The accuracy of VoVNet and HarDNet can be attributed to the One-Shot Aggregation (OSA) modules and the local feature enhancement modules, respectively while the linear stack of 36 CNN layers help Xception. The middle spectrum is occupied by *ResNet-18*, *DarkNet-19* and *DeiT-T*. ResNet and DarkNet are all the light weight architectures and DeiT enjoys the benefits of self-attention modules which helps in global context. Finally, *EfficientNet-B0*, *MobileNet-v2* and *ShuffleNet-v2*, which are sleek networks mainly designed for reducing the MAC count, achieve the least accuracy.

Amongst the heads, the single-stage keypoint-based detectors such as *NanoDet*, *FCOS*, *TTFNet* and *CenterNet* consistently achieve high accuracy. All of these detectors are keypoint estimation-based methods and do not have the hassle of defining anchor sizes based on the object sizes. *FCOS* has FPN to perform multi-scale prediction and has a centerness branch to filter out low-quality predictions. *NanoDet* incorporates PAN which enhances low-level features and uses a GIoU loss that helps refines the locations. *TTFNet* and *CenterNet* also incorporate multiple resolutions and further refine the locations. The attention modules present in *DETR* result in improved accuracy but since the backbone is still CNN, the performance is pushed to the middle spectrum. *SSD* also occupies the middle spectrum and the lower spectrum consists of *YOLO* and *ThunderNet* which are anchor-based and two-stage detectors, respectively.

**Speed**: The inference speed in Figure 12 (middle plot) shows a different trend compared to the accuracy. Amongst the backbones, the networks that were positioned in the middle spectrum in terms of "accuracy" are the fastest, namely *ResNet-18*, *DarkNet-19* and *DeiT-T*, and thus facilitate a good trade-off between accuracy and speed. Although *ShuffleNet-v2* is one of the least accurate, the inference speed is quite high owing to its pruned architecture designed for low latency. *VoVNet-39* and *HarDNet-68* which are in the highest spectrum in accuracy are positioned in the middle spectrum in terms of speed. However, *Xception* is one of the slowest due to its big linear stack of convolution layers.

Amongst the heads, *CenterNet*, *TTFNet*, and *NanoDet* are the fastest and outperform other detectors by a significant margin. *CenterNet* and *TTFNet* have no NMS bottleneck (as it uses a heatmap peak-based max-pooling NMS instead of IoU-based NMS) which helps in increasing the inference speed. *FCOS*, which has

---

[1]All the following tables and figures follow this specific order.

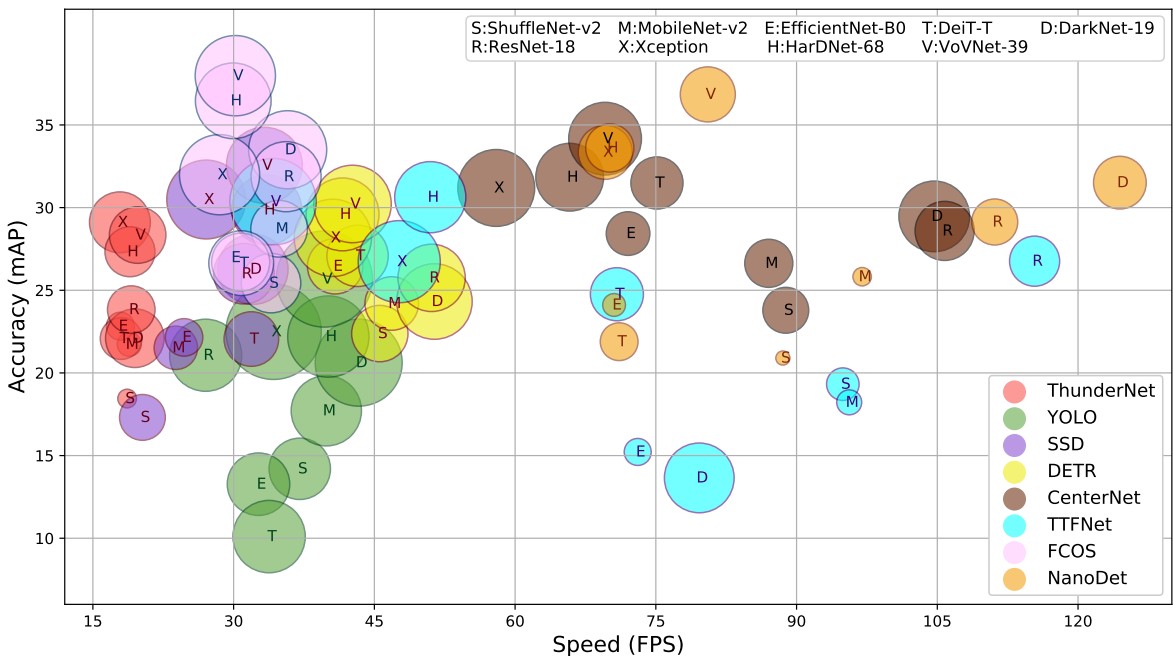

Figure 13: Speed-accuracy trade-off of all combinations of detection heads and backbones from Table 4, on COCO dataset. The size of the bubble corresponds to the number of parameters present in the model. Colors indicate detection heads and the letters represent different backbones.

the highest accuracy lies in the lowest spectrum in terms of speed owing to its heavy architecture with five feature maps and an additional centerness branch. *NanoDet* is similar to *FCOS* but has a more optimized and lightweight architecture with just three feature maps and no separate branch which results in improved inference speed. *DETR* is in the middle spectrum here, as the transformer architecture is not as hardware optimized as CNNs (Ivanov et al., 2020). *SSD* and *YOLO* also lie in the middle spectrum, achieving average speed. The two-stage based detector, *ThunderNet* is the slowest.

Further, Figure 13 shows the speed, accuracy, and parameter trade-off for all detection heads and backbones (72 combinations) on COCO dataset. The size of the each bubble indicates the number of parameters in the network. Most of the combinations result in speed range of 15 to 50 FPS, while NanoDet and CenterNet achieve higher speed for all the backbones.

**Energy and Resources**: Figure 12 (last plot) shows that amongst the backbones, the low spectrum networks consume the least amount of energy as these networks are quite small in size which is also reflected in the lower number of parameters (in Table 4). *DarkNet* proves to be quite energy efficient. Amongst the heads, the high spectrum detectors fare quite well except *FCOS* as the architecture is heavier than the other keypoint-based detectors. *NanoDet*, which was specifically designed to run on mobile hardware, consumes the least amount of energy. *SSD* and *DETR* maintain the middle spectrum in terms of energy consumption. *ThunderNet* has the proposal stage in addition to the classification and regression stage and consumes more energy compared to single stage detectors.

**Detailed Analysis:**

Table 4 provides more detailed information for eight detection heads with nine backbones across two different datasets: VOC and COCO. We report detailed information on the accuracy (mAP and F1 score), inference speed (in FPS), inference energy consumption (in kJ), and also resource information such as GMAC count and number of parameters (in million). VOC is simple in complexity and the number of classes is 20 compared

Table 4: Results of all detection models with various backbones on COCO and VOC datasets. MAC count (G), number of parameters (#P; in Million), inference energy consumption (in kilo Joules), accuracy in terms of F1 score and mAP, and inference speed in terms of FPS are reported. mAP is reported @IoU: [0.5 - 0.95] for COCO and @IoU: 0.5 for VOC. The best performance for each detector model is in bold and the overall best performance for each metric and dataset is highlighted.

| Head | Backbone | COCO | | | | | | VOC | | | | | |
|---|---|---|---|---|---|---|---|---|---|---|---|---|---|
| | | MAC (G) | #P (M) | Inf Energy (kJ) | F1 Score | mAP | Inf Speed (FPS) | MAC (G) | #P (M) | Inf Energy (kJ) | F1 Score | mAP | Inf Speed (FPS) |
| ThunderNet | ShuffleNet-v2 | **1.71** | **2.51** | **32.31** | 36.52 | 18.45 | 18.65 | **1.41** | **2.21** | **12.69** | 71.26 | 68.24 | 41.41 |
| | MobileNet-v2 | 2.71 | 4.12 | 33.83 | 40.66 | 21.73 | 18.87 | 2.41 | 3.81 | 16.23 | 74.85 | 72.14 | 41.92 |
| | EfficientNet-B0 | 2.91 | 5.60 | 38.32 | 42.06 | 22.83 | 17.93 | 2.60 | 5.29 | 24.26 | 77.11 | 74.76 | 36.67 |
| | DeiT-T | 11.89 | 12.58 | 54.21 | 40.62 | 22.07 | 18.01 | 11.58 | 12.27 | 31.41 | 74.45 | 72.01 | 35.63 |
| | DarkNet-19 | 15.57 | 24.83 | 37.93 | 41.15 | 22.09 | 19.47 | 15.26 | 24.52 | 22.81 | 78.75 | 76.59 | **44.91** |
| | ResNet-18 | 17.54 | 16.33 | 37.94 | 43.56 | 23.83 | 19.09 | 17.24 | 16.02 | 24.20 | 77.49 | 75.12 | 44.41 |
| | Xception | 25.48 | 26.78 | 61.56 | **49.89** | **29.11** | 17.88 | 25.17 | 26.47 | 36.76 | **81.64** | **79.92** | 34.05 |
| | HarDNet-68 | 23.32 | 18.04 | 42.19 | 47.75 | 27.32 | 18.95 | 23.01 | 17.73 | 27.51 | 77.70 | 75.28 | 40.18 |
| | VoVNet-39 | 38.16 | 23.11 | 53.66 | 48.99 | 28.34 | **19.77** | 37.85 | 22.80 | 30.62 | 81.00 | 79.31 | 41.21 |
| YOLO | ShuffleNet-v2 | **7.44** | **27.22** | **15.65** | 34.33 | 14.20 | 37.05 | **7.36** | **26.91** | **7.97** | 36.91 | 51.04 | **90.60** |
| | MobileNet-v2 | 10.24 | 35.70 | **15.65** | 40.16 | 17.71 | 39.88 | 10.16 | 35.39 | 12.22 | 71.45 | 71.19 | 56.75 |
| | EfficientNet-B0 | 8.28 | 28.23 | 18.76 | 32.58 | 13.27 | 32.65 | 8.20 | 27.92 | 15.78 | 41.25 | 57.26 | 75.66 |
| | DeiT-T | 17.60 | 37.85 | 30.52 | 27.06 | 10.10 | 33.78 | 17.52 | 37.54 | 28.42 | 64.35 | 63.35 | 43.86 |
| | DarkNet-19 | 22.53 | 54.63 | 17.96 | 44.07 | 20.62 | **43.33** | 22.33 | 50.66 | 18.07 | 71.14 | 73.76 | 71.12 |
| | ResNet-18 | 43.03 | 37.58 | 25.21 | 45.43 | 21.07 | 27.02 | 23.25 | 41.23 | 21.15 | 69.82 | 69.55 | 58.24 |
| | Xception | 34.75 | 65.73 | 37.38 | 47.74 | 22.49 | 34.26 | 34.67 | 65.43 | 30.20 | 73.26 | 75.88 | 43.80 |
| | HarDNet-68 | 30.26 | 47.69 | 24.40 | 46.90 | 22.20 | 40.09 | 30.18 | 47.39 | 25.52 | 75.04 | 76.86 | 48.70 |
| | VoVNet-39 | 54.79 | 66.78 | 33.31 | **51.02** | **25.68** | 39.67 | 44.99 | 52.43 | 24.67 | **76.03** | **78.61** | 53.94 |
| SSD | ShuffleNet-v2 | 4.24 | 15.14 | **24.16** | 37.45 | 17.31 | 20.28 | **1.92** | 8.21 | **14.55** | 72.35 | 67.38 | 22.40 |
| | MobileNet-v2 | 4.30 | 13.60 | 26.22 | 43.97 | 21.52 | 23.84 | 2.51 | **6.03** | 18.08 | 78.56 | 72.54 | 48.53 |
| | EfficientNet-B0 | **3.52** | **10.16** | 26.68 | 31.77 | 22.15 | 24.72 | 2.46 | 6.46 | 19.86 | 79.54 | 73.17 | 45.92 |
| | DeiT-T | 15.68 | 21.53 | 35.39 | 48.98 | 22.05 | 31.89 | 12.78 | 14.41 | 21.83 | 79.31 | 74.54 | 52.92 |
| | DarkNet-19 | 21.44 | 35.63 | 31.54 | 50.72 | 26.27 | 32.06 | 16.56 | 27.06 | 16.52 | 83.17 | 77.84 | 71.13 |
| | ResNet-18 | 21.35 | 26.47 | 31.81 | 50.87 | 26.00 | 31.10 | 17.95 | 18.45 | 17.22 | 82.00 | 76.47 | **74.00** |
| | Xception | 33.55 | 44.93 | 47.56 | 55.59 | 30.49 | 27.08 | 27.04 | 30.96 | 27.03 | 82.59 | 78.86 | 50.05 |
| | HarDNet-68 | 30.85 | 35.49 | 35.34 | 55.79 | 29.86 | **33.53** | 24.91 | 24.99 | 21.07 | 85.37 | 80.41 | 60.12 |
| | VoVNet-39 | 48.66 | 41.70 | 40.77 | **58.88** | **32.55** | 33.28 | 40.59 | 30.37 | 21.20 | **85.80** | **81.57** | 65.29 |
| DETR | ShuffleNet-v2 | 4.17 | 22.98 | **16.93** | 47.27 | 22.38 | 45.58 | 4.16 | 22.97 | **11.90** | 77.52 | 69.72 | 46.97 |
| | MobileNet-v2 | 5.21 | **20.75** | 18.88 | 49.81 | 24.20 | 46.85 | 5.20 | **20.73** | 14.91 | 78.15 | 70.78 | 48.24 |
| | EfficientNet-B0 | 5.49 | 22.21 | 26.56 | 53.68 | 26.45 | 40.82 | 5.48 | 22.20 | 23.17 | 81.95 | 75.31 | 42.76 |
| | DeiT-T | 14.35 | 27.21 | 30.64 | 53.09 | 27.08 | 43.21 | 14.34 | 27.20 | 27.09 | 80.11 | 73.07 | 43.95 |
| | DarkNet-19 | 17.92 | 41.29 | 25.09 | 49.72 | 24.32 | **51.40** | 17.91 | 41.28 | 21.59 | 81.00 | 74.29 | **51.58** |
| | ResNet-18 | 19.90 | 32.79 | 26.32 | 51.68 | 25.75 | 51.10 | 19.90 | 32.77 | 21.32 | 79.68 | 72.68 | 50.88 |
| | Xception | 27.65 | 43.20 | 36.46 | 54.90 | 28.17 | 40.57 | 27.64 | 43.19 | 31.51 | 83.13 | 77.32 | 41.39 |
| | HarDNet-68 | 25.78 | 38.41 | 30.86 | 56.44 | 29.59 | 41.61 | 25.77 | 38.40 | 27.57 | 83.70 | 77.44 | 41.64 |
| | VoVNet-39 | 40.53 | 43.45 | 34.23 | **57.88** | **30.22** | 42.67 | 40.52 | 43.43 | 28.97 | **84.57** | **78.67** | 44.08 |
| CenterNet | ShuffleNet-v2 | 12.04 | 15.20 | 19.31 | 48.20 | 23.79 | 88.85 | 11.92 | 15.19 | 16.48 | 76.36 | 68.42 | 79.71 |
| | MobileNet-v2 | 12.90 | 16.71 | 20.84 | 52.21 | 26.63 | 87.04 | 12.77 | 16.70 | 15.70 | 80.11 | 72.84 | 89.44 |
| | EfficientNet-B0 | 13.09 | **13.76** | 23.61 | 54.29 | 28.43 | 72.04 | 12.97 | **13.76** | 18.69 | 81.05 | 73.14 | 73.65 |
| | DeiT-T | 21.89 | 19.65 | 24.12 | 58.05 | 31.49 | 75.12 | 21.77 | 19.64 | 18.48 | 81.03 | 75.24 | 75.14 |
| | DarkNet-19 | 25.65 | 36.08 | 19.35 | 55.62 | 29.47 | 104.67 | 25.53 | 36.07 | 15.32 | 82.66 | 76.47 | 103.19 |
| | ResNet-18 | 27.63 | 25.22 | **19.05** | 54.49 | 28.59 | **105.78** | 27.50 | 25.21 | **14.69** | 81.68 | 75.18 | **104.98** |
| | Xception | 35.50 | 42.70 | 29.33 | 57.35 | 31.19 | 57.99 | 35.38 | 42.69 | 24.25 | 82.73 | 77.30 | 57.07 |
| | HarDNet-68 | 33.55 | 33.20 | 25.97 | 58.19 | 31.84 | 65.82 | 33.42 | 33.19 | 20.81 | 82.69 | 77.49 | 65.70 |
| | VoVNet-39 | 48.35 | 38.23 | 26.25 | **60.92** | **34.17** | 69.59 | 48.23 | 38.23 | 22.05 | **84.88** | **79.73** | 64.98 |
| TTFNet | ShuffleNet-v2 | 5.55 | 7.66 | 15.33 | 44.15 | 19.32 | 94.93 | 5.44 | 7.65 | 12.08 | 75.93 | 68.37 | 94.49 |
| | MobileNet-v2 | **2.72** | **4.50** | **15.14** | 41.74 | 18.22 | 95.61 | **2.68** | **4.50** | **11.89** | 78.37 | 71.56 | 97.45 |
| | EfficientNet-B0 | 3.12 | 5.26 | 19.17 | 37.71 | 15.22 | 73.07 | 3.08 | 5.26 | 15.90 | 65.32 | 60.97 | 74.23 |
| | DeiT-T | 21.14 | 20.35 | 24.85 | 50.37 | 24.76 | 70.85 | 21.09 | 20.34 | 19.52 | 79.67 | 74.41 | 70.57 |
| | DarkNet-19 | 46.77 | 35.16 | 23.40 | 34.31 | 13.65 | 79.63 | 46.52 | 35.14 | 18.16 | 83.62 | 77.62 | 79.80 |
| | ResNet-18 | 24.81 | 18.15 | 18.05 | 53.27 | 26.76 | **115.37** | 24.68 | 18.14 | 13.57 | 81.62 | 75.31 | **113.08** |
| | Xception | 59.15 | 48.72 | 34.43 | 54.73 | 26.73 | 47.65 | 58.89 | 48.70 | 29.49 | 81.43 | 74.70 | 46.75 |
| | HarDNet-68 | 65.57 | 36.64 | 32.27 | 57.52 | **30.63** | 50.96 | 65.26 | 36.62 | 26.72 | 85.17 | 80.15 | 50.00 |
| | VoVNet-39 | 160.17 | 53.98 | 46.65 | **57.77** | 30.36 | 34.20 | 159.66 | 53.94 | 42.60 | **86.05** | **81.20** | 33.11 |
| | ShuffleNet-v2 | **106.58** | 25.55 | 43.45 | 50.73 | 25.43 | 33.99 | **105.07** | 25.27 | 37.98 | 79.95 | 71.41 | 33.67 |
| | MobileNet-v2 | 107.24 | **23.28** | **42.96** | 54.71 | 28.72 | 34.85 | 105.73 | **23.00** | **37.27** | 83.56 | 76.06 | 34.88 |
| | EfficientNet-B0 | 107.77 | 24.38 | 47.93 | 52.99 | 26.70 | 30.75 | 106.26 | 24.10 | 43.22 | 83.33 | 75.74 | 30.46 |
| | DeiT-T | 117.39 | 30.40 | 48.64 | 54.02 | 26.64 | 30.81 | 115.88 | 30.12 | 43.46 | 84.22 | 77.19 | 30.72 |

| | | | | | | | | | | | | |
|---|---|---|---|---|---|---|---|---|---|---|---|---|
| **FCOS** | DarkNet-19 | 120.23 | 44.09 | 43.05 | 59.82 | 33.50 | **35.76** | 118.73 | 43.81 | 37.33 | 86.12 | 79.61 | **35.78** |
| | ResNet-18 | 122.34 | 35.51 | 43.80 | 58.32 | 31.87 | 35.59 | 120.83 | 35.24 | 37.36 | 84.85 | 77.98 | 35.58 |
| | Xception | 130.70 | 46.50 | 51.94 | 57.75 | 31.99 | 28.50 | 129.19 | 46.22 | 46.53 | 86.26 | 80.08 | 28.44 |
| | HarDNet-68 | 128.76 | 41.43 | 49.16 | 63.67 | 36.45 | 29.96 | 127.25 | 41.16 | 43.56 | 87.86 | 81.59 | 30.03 |
| | VoVNet-39 | 144.04 | 46.63 | 49.65 | **64.63** | **37.98** | 30.18 | 142.53 | 46.36 | 43.73 | **88.01** | **81.79** | 30.37 |
| **NanoDet** | ShuffleNet-v2 | **1.04** | **1.43** | **10.66** | 45.66 | 20.90 | 88.54 | **1.00** | **1.41** | **7.28** | 75.88 | 66.81 | 87.45 |
| | MobileNet-v2 | 1.89 | 2.45 | 13.79 | 52.09 | 25.82 | 96.99 | 1.86 | 2.44 | 10.80 | 81.92 | 74.67 | 98.91 |
| | EfficientNet-B0 | 2.18 | 3.74 | 18.39 | 50.48 | 24.11 | 70.55 | 2.15 | 3.72 | 15.49 | 82.69 | 75.45 | 70.70 |
| | DeiT-T | 11.71 | 10.31 | 22.99 | 47.91 | 21.89 | 71.09 | 11.68 | 10.29 | 18.84 | 81.47 | 75.11 | 71.64 |
| | DarkNet-19 | 14.76 | 20.09 | 14.45 | 59.27 | 31.51 | **124.46** | 14.72 | 20.07 | 11.55 | 84.32 | 79.13 | **128.48** |
| | ResNet-18 | 16.83 | 15.32 | 16.41 | 55.80 | 29.12 | 111.12 | 16.79 | 15.30 | 13.21 | 82.87 | 76.15 | 112.33 |
| | Xception | 24.48 | 21.19 | 24.84 | 61.22 | 33.35 | 69.61 | 24.45 | 21.18 | 19.59 | 85.59 | 79.94 | 69.99 |
| | HarDNet-68 | 22.64 | 16.83 | 21.21 | 61.00 | 33.61 | 70.09 | 22.61 | 16.81 | 18.25 | 85.91 | 80.35 | 71.42 |
| | VoVNet-39 | 37.53 | 21.89 | 22.26 | **64.84** | **36.85** | 80.54 | 37.50 | 21.87 | 17.32 | **86.09** | **81.67** | 80.58 |

to COCO which has 80 classes. The accuracy (both mAP and F1 score) is at a higher scale in VOC compared to COCO, but networks show similar trends in the case of both datasets. *FCOS + VoVNet-39* pair has the highest accuracy while *NanoDet+DarkNet-19* pair has the highest inference speed in both the datasets. Among the mid-spectrum backbones, *DeiT-T* displays lower MAC count. The resources across both datasets are similar except in *SSD* where the number of parameters increases (in some cases around 2×) upon changing the dataset from VOC to COCO. As *SSD* uses six feature maps with separate anchor boxes for each, there is a resource overhead when the number of classes increases. The effect is less amplified in anchor-free designs.

Overall, among the backbones, the high spectrum networks *HarDNet-68* and *VoVNet-39* perform well across all metrics but *Xception* falls short in all the metrics except accuracy. The middle spectrum comprising of *ResNet-18*, *DarkNet-19* and *DeiT-T* offer a good balance between accuracy, speed, and resources. *DeiT-T*, being convolution free is the most resource friendly with the least MAC count. Among detection heads, *NanoDet* achieves high accuracy and speed while also being computationally efficient. *CenterNet* and *TTFNet* also offer a good balance while *TTFNet* facilitates faster training times. *DETR* (likewise, *NanoDet*) displays low MAC count when paired with the lighter backbones. Our detailed results dispense an atlas for choosing a network based on specific requirements. Furthermore, it also acts as a comparative baseline while designing and testing new architectures.

To further demonstrate the importance of evaluating the networks on different metrics, we report the Pearson correlation between all the metrics in Figure 14 for the results of both COCO and VOC datasets in Table 4. As the plot is symmetric, only the lower triangle is displayed. As seen, accuracy is positively correlated with all the other metrics with GMAC being the highest correlated after F1 score. Speed is only highly negatively correlated with inference energy. Energy has the highest positive correlation with GMAC, indicating that networks with more MAC operations tend to consume more energy.

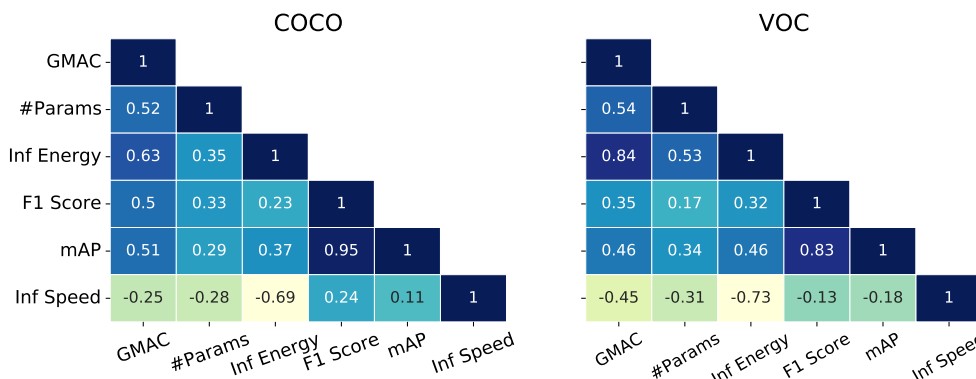

Figure 14: Correlation between different parameters reported in Table 4 for all heads and backbones.

**Small objects**

| | ShuffleNet-v2 | MobileNet-v2 | EfficientNet-B0 | DeiT-T | DarkNet-19 | ResNet-18 | Xception | HarDNet-68 | VoVNet-39 |
|---|---|---|---|---|---|---|---|---|---|
| ThunderNet | 3.5 | 4.8 | 4.7 | 4.1 | 5.1 | 6.2 | 8.7 | 7.8 | 8.6 |
| YOLO | 2.6 | 3.8 | 2.9 | 2.0 | 5.3 | 6.1 | 5.8 | 5.6 | 7.6 |
| SSD | 3.0 | 4.3 | 4.7 | 6.8 | 6.9 | 7.3 | 8.4 | 9.9 | 12.3 |
| DETR | 4.3 | 5.1 | 5.9 | 6.9 | 5.4 | 6.5 | 6.9 | 8.7 | 8.4 |
| CenterNet | 6.5 | 8.2 | 8.7 | 11.0 | 10.5 | 10.2 | 10.3 | 10.5 | 13.5 |
| TTFNet | 7.4 | 7.1 | 5.6 | 9.6 | 7.7 | 11.9 | 11.6 | 15.5 | 16.9 |
| FCOS | 10.8 | 13.6 | 12.7 | 12.7 | 15.8 | 15.2 | 12.8 | 18.3 | 20.8 |
| NanoDet | 8.4 | 11.9 | 10.6 | 10.3 | 15.1 | 13.4 | 16.5 | 16.6 | 19.9 |

**Medium objects**

| | ShuffleNet-v2 | MobileNet-v2 | EfficientNet-B0 | DeiT-T | DarkNet-19 | ResNet-18 | Xception | HarDNet-68 | VoVNet-39 |
|---|---|---|---|---|---|---|---|---|---|
| ThunderNet | 15.4 | 19.3 | 20.4 | 18.7 | 20.0 | 22.1 | 29.2 | 26.4 | 28.4 |
| YOLO | 14.3 | 18.2 | 13.5 | 9.8 | 21.5 | 23.4 | 24.0 | 24.2 | 28.3 |
| SSD | 16.8 | 22.6 | 23.5 | 22.3 | 28.7 | 29.4 | 36.6 | 34.0 | 36.8 |
| DETR | 20.5 | 23.0 | 25.7 | 26.4 | 23.2 | 25.1 | 27.4 | 30.4 | 30.8 |
| CenterNet | 25.1 | 28.9 | 31.8 | 34.0 | 31.5 | 30.6 | 33.9 | 35.0 | 37.9 |
| TTFNet | 19.7 | 18.4 | 15.3 | 24.5 | 14.6 | 27.8 | 28.1 | 32.7 | 32.6 |
| FCOS | 24.3 | 27.6 | 26.2 | 27.8 | 34.0 | 32.1 | 31.6 | 38.7 | 39.9 |
| NanoDet | 20.1 | 25.8 | 23.9 | 20.9 | 32.9 | 29.8 | 35.1 | 35.4 | 38.6 |

**Large objects**

| | ShuffleNet-v2 | MobileNet-v2 | EfficientNet-B0 | DeiT-T | DarkNet-19 | ResNet-18 | Xception | HarDNet-68 | VoVNet-39 |
|---|---|---|---|---|---|---|---|---|---|
| ThunderNet | 34.8 | 39.3 | 42.7 | 42.1 | 39.5 | 42.7 | 51.7 | 47.8 | 49.1 |
| YOLO | 23.6 | 29.2 | 21.4 | 16.3 | 34.2 | 31.9 | 37.7 | 35.5 | 41.0 |
| SSD | 33.2 | 39.1 | 40.4 | 38.4 | 44.8 | 43.3 | 51.4 | 47.1 | 51.6 |
| DETR | 41.9 | 44.2 | 48.8 | 48.4 | 43.9 | 45.5 | 51.5 | 52.2 | 52.9 |
| CenterNet | 40.7 | 44.0 | 46.7 | 51.5 | 48.8 | 46.2 | 52.6 | 52.8 | 54.4 |
| TTFNet | 29.5 | 26.5 | 23.4 | 40.2 | 17.1 | 39.2 | 41.8 | 42.5 | 41.2 |
| FCOS | 41.8 | 46.2 | 42.3 | 40.2 | 52.9 | 50.3 | 54.2 | 54.4 | 56.2 |
| NanoDet | 32.5 | 38.5 | 36.6 | 33.4 | 46.8 | 43.3 | 50.9 | 49.1 | 53.9 |

Figure 15: Accuracy (mAP) of all detection networks in conjunction with various backbones reported on different object sizes (small, medium and large) @IoU: [0.5-0.95] on COCO dataset. Darker shades represent higher accuracy values.

The qualitative results of each detector on the COCO dataset are presented in Figure 23 in Appendix.

# 9 Decoupling Different Effects

The accuracy of the networks can depend on multiple factors, thus in this section, we provide further analyses to decouple these variables. This provides detailed insights into the capabilities of the network that are often overlooked while providing benchmarks. We study the network's performance on various object sizes viz. small, medium, and large. We also show the gain/loss in accuracy and speed when using input images of different resolutions. The role of the confidence threshold parameter while reporting performances is explained with examples. For anchor-based approaches, we show the sensitivity of the networks to different anchor sizes. Finally, we decouple an architectural element, namely the Deformable Convolution (DCN) layer in the networks to understand the accuracy and speed trade-off. This section provides a rare outlook on the effect of many understated variables on object detectors' performance to provide a holistic picture that helps in both deployment and design of architectures.

## 9.1 Effect of Object Size

Detection of small objects is a challenging problem for most object detectors (Liu et al., 2021). To show the performance of networks across objects of different sizes, we compare the accuracy of the backbones and detection heads across the three different sizes, i.e. small, medium and large. Figure 15 shows that all backbones and detection heads struggle to obtain high accuracy for smaller objects. In this context, *TTFNet*, *NanoDet*, and *FCOS* outperform the rest, mostly when complimented by the best performing heavy backbones (see Table 4) such as *HarDNet-68* or *VoVNet-39*. The higher resolution feature maps from the heavier backbones combined with the FPN/PAN in these detection heads work in their favor. The accuracy of medium and large objects are considerably better. Among the heads, *FCOS* and *NanoDet* perform better overall for objects of all sizes. *TTFNet*, *CenterNet*, and *SSD*, positioned in the middle spectrum, with faster backbones can be good choices for applications which require higher inference speed. The consistent performance of *FCOS* can be attributed to its heavier architecture owing to the use of five feature maps of varying resolutions.

For further analyses, we consider all the detection heads with *HarDNet-68* backbone as it provides the best compromise on a more complex dataset, namely COCO.

## 9.2 Effect of Input Image Size

The resolution of the input image used for training plays a major role in the final accuracy. The image resolution used in all prior experiments is 512×512. To analyze the accuracy and speed trade-off of other input resolutions, we use different image sizes for training, including 256, 384, 512, and 736. The image sizes are chosen as an "even multiple of 16", a common feature map used by multiple detection heads.

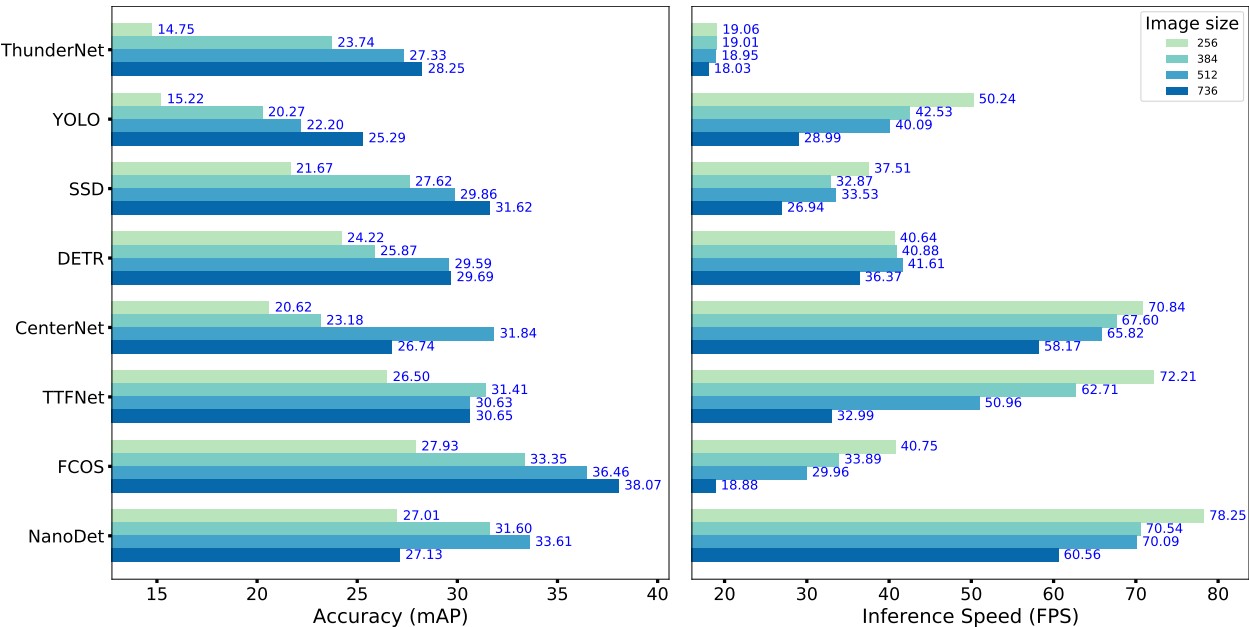

Figure 16: Effect of image sizes on accuracy and inference speed of detection models with HarDNet-68 backbone on COCO dataset. As image size increases, the gain in accuracy is masked by the decline in speed.

Figure 16 demonstrates that the accuracy across the detection heads follow a trend of "diminishing returns". In most cases, there is a significant accuracy jump from 256 to 384 image resolution. However, the gain decreases as the image size increases further with the least gain observed when the image size changes from 512 to 736 (also, in some cases it reduces the accuracy). Additionally, we observe that the gain in accuracy for higher image sizes is masked by a bigger decline in speed. For instance, a ∼4.4% accuracy gain in *FCOS*, from resolution 512 to 736, is masked by the 37% decrease in speed and hence not an efficient choice to switch to higher resolutions. The speed decline with increasing resolution is more pronounced for *YOLO*, *TTFNet*, and *FCOS* as the number of operations in multi-resolution feature concatenation in *YOLO* and in FPN (in the others) increases with image size. *DETR* which employs attention blocks to capture the global context of the images and *ThunderNet* with separate region proposals for each sample respectively are less sensitive to varying image sizes.

### 9.3 Effect of anchor box size

Anchor sizes and aspect ratios need to be in line with the size of the objects present in the dataset and hence are an important parameter in anchor-based networks. The anchor sizes also need to be known a priori and hence is difficult to adapt to new datasets. Out of the eight detection heads in our study, *SSD*, *YOLO*, and *Thundernet* use anchor-based approach for detection. To analyze the impact of anchors on detection performance, we conduct experiments on all the three anchor-based detection heads with varying anchor box sizes which are defined by their respective width and height. The aim is to add some offsets to anchors' width and height and analyze the change in speed and accuracy of the networks. Instead of linearly increasing/decreasing the anchor dimensions, we sample the offsets from a Gaussian distribution with varying sigma, and add this to the original anchor width and height to create the modified anchor dimensions. Moreover, the modified anchors are kept constant throughout that specific experiment. The width and height of the original anchors (from the original architecture of these networks) are considered as the baseline.

Table 5 shows the change in accuracy and inference speed with modified anchors in percentage w.r.t to the baseline (first row). We observe that the change in accuracy at anchors of different sizes follows a random pattern and there is no correlation between accuracy and speed. *ThunderNet* is less sensitive to changes in

Table 5: Effect of anchor box sizes on accuracy (mAP) and inference speed (FPS) for the three anchor-based detection networks with HarDNet-68 backbone on COCO dataset (@IoU: [0.5, 0.95]). Numbers are reported in terms of percentage change w.r.t the original anchor sizes. The best performance for each detector is in bold. There is no common pattern and the performance changes in a non-deterministic way with respect to the anchor box sizes.

| Anchor size | ThunderNet | | YOLO | | SSD | |
|---|---|---|---|---|---|---|
| | mAP | FPS | mAP | FPS | mAP | FPS |
| **Orig.** | 27.32 | 18.65 | 22.20 | 40.09 | **29.86** | 33.53 |
| 0.2 $\sigma$ | +0.26 | +4.00 | +3.38 | -2.51 | -0.21 | -2.46 |
| 0.3 $\sigma$ | +1.06 | **+4.90** | +2.32 | -6.70 | -4.29 | **+5.85** |
| 0.5 $\sigma$ | **+1.79** | +4.72 | **+3.69** | **+2.38** | -9.76 | -2.09 |

Table 6: Effect of confidence threshold on performance and inference speed (FPS) on detectors with HarDNet-68 backbone on COCO dataset (@IoU: [0.5-0.95]). The results show that a lower threshold yields higher accuracy while a higher threshold yields higher speed.

| Head | Threshold 0.01 | | | Threshold 0.4 | | |
|---|---|---|---|---|---|---|
| | mAR | mAP | FPS | mAR | mAP | FPS |
| YOLO | **53.74** | **22.20** | 40.09 | 30.51 | 17.15 | **68.52** |
| SSD | **66.26** | **29.86** | 33.53 | 39.05 | 24.32 | **49.95** |
| FCOS | **76.74** | **36.45** | 29.96 | 43.97 | 29.24 | **31.12** |
| NanoDet | **74.20** | **33.61** | 70.09 | 44.73 | 28.26 | **72.15** |

anchor box sizes in terms of accuracy while its inference speed boosts constantly. *SSD* is very sensitive to these changes as its accuracy decreases for all the three sets of anchor box sizes while the speed increases for one of them. The changes improve the accuracy of *YOLO* but do not affect the inference speed uniformly. The non-deterministic way the anchor box sizes affect the detection proves that modifying anchors to improve detection is not a simple and straightforward task.

### 9.4 Effect of Confidence Thresholds

Object detectors produce many boxes and a threshold is used to filter out the redundant and low confident predictions. Varying this threshold affects precision and recall. Hence, the confidence threshold plays a vital role in calculating accuracy and inference speed. There has been a discrepancy in reproducing the results from prior object detection literature as such parameters are not explicitly mentioned. Using different thresholds shows a significant difference in accuracy and inference numbers. To exhibit this non-uniformity, we analyze the networks with different confidence thresholds and report the performance in terms of mAR and mAP, and speed for each.

*ThunderNet* has region-based proposals, and utilizes a Soft-NMS and the scores are decayed instead of fixing a hard threshold, hence this parameter does not affect the results. *CenterNet* and *TTFNet* use maxpool to select the predictions, and *DETR* removes the traditional detection modules and hence do not use this threshold. Thus, only *YOLO*, *SSD*, *FCOS*, and *NanoDet* are considered for this study. Table 6 shows the dip in accuracy while using a higher threshold and the significant increase in speed while using the lower threshold. For example, the mAP of *YOLO* dips by ∼22% while the speed increase by ∼71% by changing the threshold from 0.01 to 0.4. For applications such as medical imaging, where the recall metric is more relevant to eliminating false negative cases, the decline in accuracy is even more prominent and hence can have a higher impact. This analysis is an attempt to encourage transparency and fairness in comparison to all future works, which will help the reproducibility.

Table 7: Effect of DCN layers on accuracy and efficiency metrics. The detectors with HarDNet-68 backbone are trained and tested on COCO and BDD datasets (@IoU: [0.5-0.95]). The gain in accuracy using DCN comes at the cost of lower speed and higher resource requirements and energy consumption.

| Head | DCN Layer | COCO | | | | BDD | | | |
|---|---|---|---|---|---|---|---|---|---|
| | | mAP | Inf Speed (FPS) | #Params (M) | Energy (kJ) | mAP | Inf Speed (FPS) | #Params (M) | Energy (kJ) |
| CenterNet | ✓ | **31.84** | 65.82 | 33.20 | 25.97 | **22.12** | 66.57 | 33.19 | 43.69 |
| | ✗ | 29.67 | **71.77** | **32.76** | **24.72** | 21.72 | **72.43** | **32.76** | **38.58** |
| TTFNet | ✓ | **30.63** | 50.96 | 36.64 | 32.27 | **22.01** | 55.81 | 36.22 | 51.81 |
| | ✗ | 29.13 | **56.10** | **36.16** | **30.34** | 20.98 | **56.98** | **36.14** | **51.13** |

### 9.5 Effect of Deformable Convolution Layer

Dai et al. (2017) introduced Deformable Convolution (DCN) layer that help in detecting objects with geometrical deformations. Conventional convolutions use a fixed rectangular grid on the image, based on defined kernel size. In DCN, each grid point can be moved by a learnable offset, i.e. the grid is deformable. DCN benchmarks focus mainly on accuracy gains and not on the other metrics. To obtain more information on speed and resource requirements, we analyze the effect of DCN layers on the two detectors that use DCN in their original proposed architecture, i.e. *CenterNet* and *TTFNet*.

Table 7 provides accuracy, speed, number of parameters, and energy consumption of the two aforementioned detectors with and without DCN layers. The results are reported on two datasets, *COCO* and *BDD*. On the BDD dataset, replacing the DCN layers with standard convolution layers in *CenterNet* results in 1.8% decrease in accuracy while the speed increase by more than 8%. The usage of DCN layers also increases the number of parameters and leads to 13% higher energy consumption. On the COCO dataset, changing DCN layers from *TTFNet* to standard convolution layers decreases the accuracy by less than 5% while the speed increase by 10% and energy consumption improves by around 6%. These results demonstrate that there is an inherent accuracy, speed, and resource requirement trade-off when using DCN layers.

Overall, this section acts as a guideline for new design considerations and also for selecting existing architectures based on the application criteria.

## 10 Calibration of Object Detectors

Many applications, especially safety-critical ones, need the detection networks to be highly accurate and reliable. Detectors must not only be accurate, but should also indicate when they are likely to be incorrect. Model calibration provides insight into model uncertainty, which can be afterwards communicated to end-users or assisting in further processing of the model outputs. It refers to the metric that the probability associated with one prediction reflects the overall accuracy likelihood. Most works focus on improving only predictive accuracy of the networks but it is essential to have a model that is also well calibrated. Large and accurate networks tend to be overconfident (Guo et al., 2017) and miscalibrated. Hence, there is an urgent need to revisit and measure the calibration of the SOTA detectors to get a complete assessment.

Most of the work on calibration has been concentrated on classification domain but Kuppers et al. (2020) includes the bounding box predictions along with the classification labels to evaluate the overall calibration of the detectors. Expected Calibration Error (ECE) (Naeini et al., 2015) is one of the common metric devised to measure calibration and measures the difference in expectation between predicted confidence and accuracy. In classification domain, this score signifies the deviation of the classification accuracy from the estimated confidence. The Detection ECE (D-ECE) (Kuppers et al., 2020) measures the deviation of the observed average precision (AP) w.r.t to both classification and the bounding box properties. The confidence space as well as the bounding box space is partitioned into equal bins and D-ECE is calculated by iterating over all the bins and accumulating the difference between AP and confidence in each bin. One dimensional case considers only the confidence but we use the multidimensional D-ECE case that combines all the factors: $p$, $cx$, $cy$, $w$, $h$ representing the class probability, center coordinates, width, and height of the predictions, respectively.

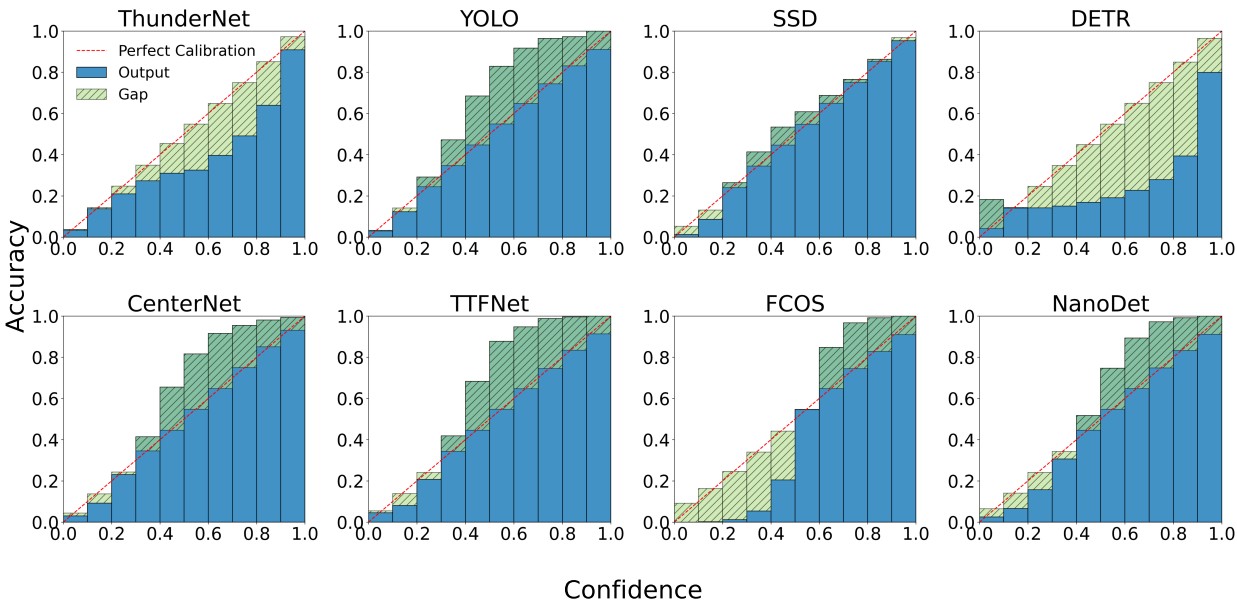

Figure 17: Reliability diagrams (based on classification prediction) of detection networks with HarDNet-68 backbone trained and tested on COCO dataset. Green shaded areas indicate the error compared to perfect calibration - darker shade of green indicates the underconfident predictions whereas the lighter shade indicates the overconfident predictions. The better the calibration, the more reliable the predictions are. *SSD* is relatively well calibrated. In general, the single-stage detectors are more underconfident while the two-stage *ThunderNet* and *DETR* are very overconfident.

Table 8: Calibration error for detection networks with HarDNet-68 backbone trained and tested on COCO dataset. Each row shows the error for a specific dimension: classification confidence only (ECE) and confidence + bounding box (D-ECE). The lowest calibration errors across all detectors are highlighted.

| Head | ThunderNet | YOLO | SSD | DETR | CenterNet | TTFNet | FCOS | NanoDet |
|---|---|---|---|---|---|---|---|---|
| **ECE** (%) | 8.09 | 5.88 | 4.07 | 18.46 | 4.31 | 6.17 | 21.56 | 6.48 |
| **D-ECE** (%) | 14.94 | 10.92 | 7.60 | 23.48 | 8.04 | 10.62 | 22.20 | 9.02 |

Reliability diagrams (DeGroot & Fienberg, 1983) are used to visually represent model calibration, where accuracy is plotted as a function of confidence.

Reliability scores and diagrams are provided in Table 8 and Figure 17, respectively. In the reliability diagrams, the diagonal represents the perfect calibration and the green shades represent the gap in calibration. Among anchor-based detectors, *SSD* is quite well calibrated while *YOLO* is more underconfident. All the keypoint-based methods (last row in the diagram) are leaning more towards being underconfident and are more cautious about their predictions and hence can be more favorable for safety-critical applications. But, the transformer-based (*DETR*) and two-stage based (*ThunderNet*) detectors are highly overconfident and can be undesirable in safety-critical applications. The calibration error increases when the localization is also included (as reflected in D-ECE). The reliability diagrams for the VOC dataset are provided in Figure 20. We further provide the reliability diagrams of all detectors trained on COCO dataset and tested on Corrupted COCO dataset in Figure 21. Very similar pattern holds for OOD calibration analysis variant. For more details, see Section A.4 in Appendix.

We note that there are several calibration solutions in the domain of classification such as histogram binning (Zadrozny & Elkan, 2001), logistic calibration/Platt scaling (Platt et al., 1999), temperature scaling (Guo et al., 2017), and beta calibration (Kull et al., 2017). However, applying these to object detection might not be as effective, thus other works (Neumann et al., 2018; Kuppers et al., 2020) have been proposed to procure

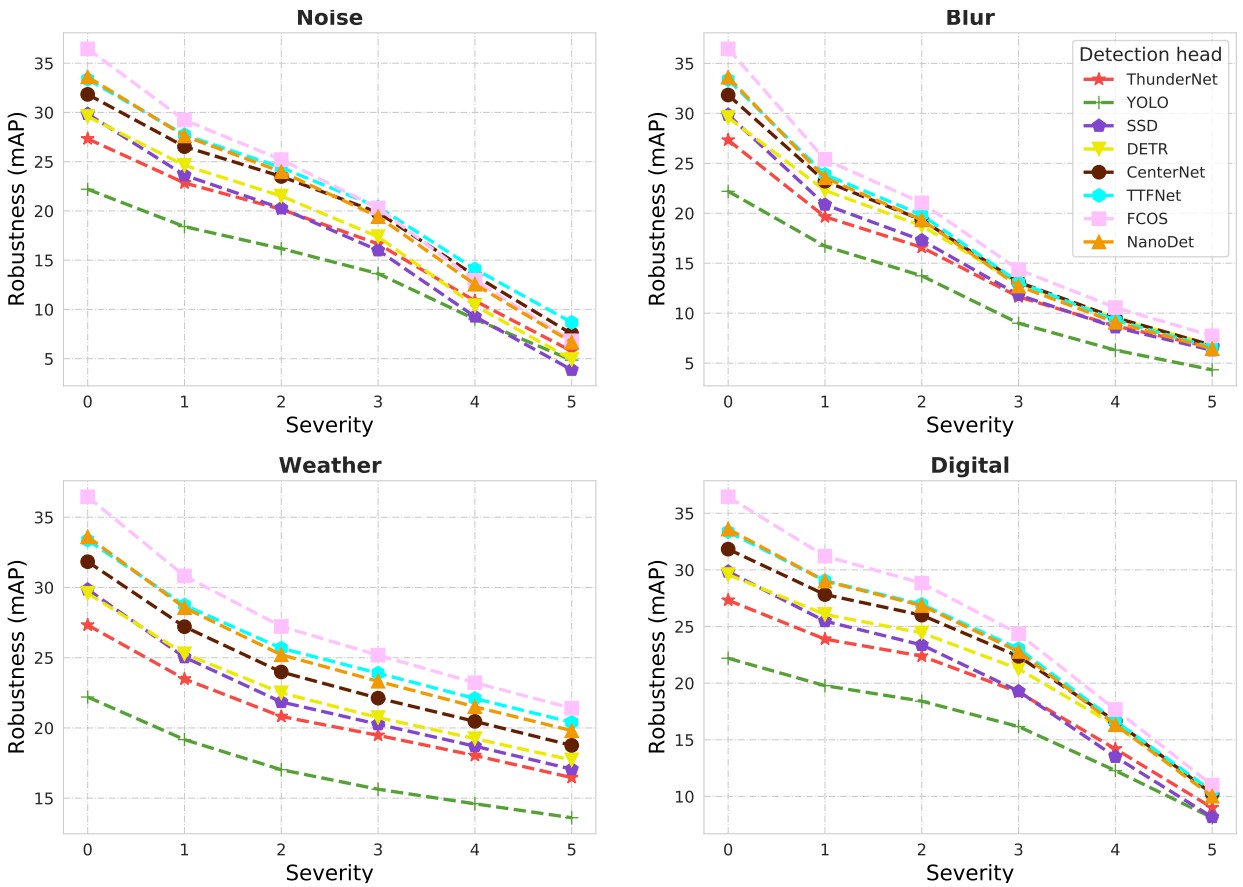

Figure 18: Robustness of detection networks with HarDNet-68 backbone trained on COCO and tested on Corrupted COCO dataset. Results are shown on four categories of corruptions: Noise, Blur, Weather, and Digital.

well-calibrated estimates for object detection specifically. In this study, we concentrate on comparing the reliability of different detectors as is, and do not delve deep into the solutions to improve their calibration.

## 11 Natural Robustness

Real-time object detection applications such as autonomous driving put huge emphasis on safety and precision. Object detectors used in such applications need to be consistent in their predictions and robust to various factors such as the changing weather conditions, lighting, and various other imaging effects. Public datasets do not have sufficient coverage of all these effects and hence we simulate them by adding different corruptions on them. As explained in Section 7.1, the Corrupted COCO dataset is created with 15 different corruptions.

Figure 18 shows the results of each head on the four categories of corruptions: Noise, Blur, Weather and Digital effects. The accuracy values are the mean of different corruptions in that particular category. The severity level 0 is the performance of the networks on the original data. The performance of all the networks deteriorate on all the corruptions and it declines faster as the severity increases. On Noise, Blur and digital effects, the networks show relatively steeper decline in performance compared to the weather category. For all corruption categories, *FCOS* is the most robust while *YOLO* is the least robust. The top, middle and low spectrum of detectors in terms of accuracy on IID data (Figure 12), still holds good on the OOD setting too Miller et al. (2021). FCOS which proved to be the most accurate in the IID test set, continues this performance even on the challenging OOD setting, i.e. for natural corrupted data. And in general, the

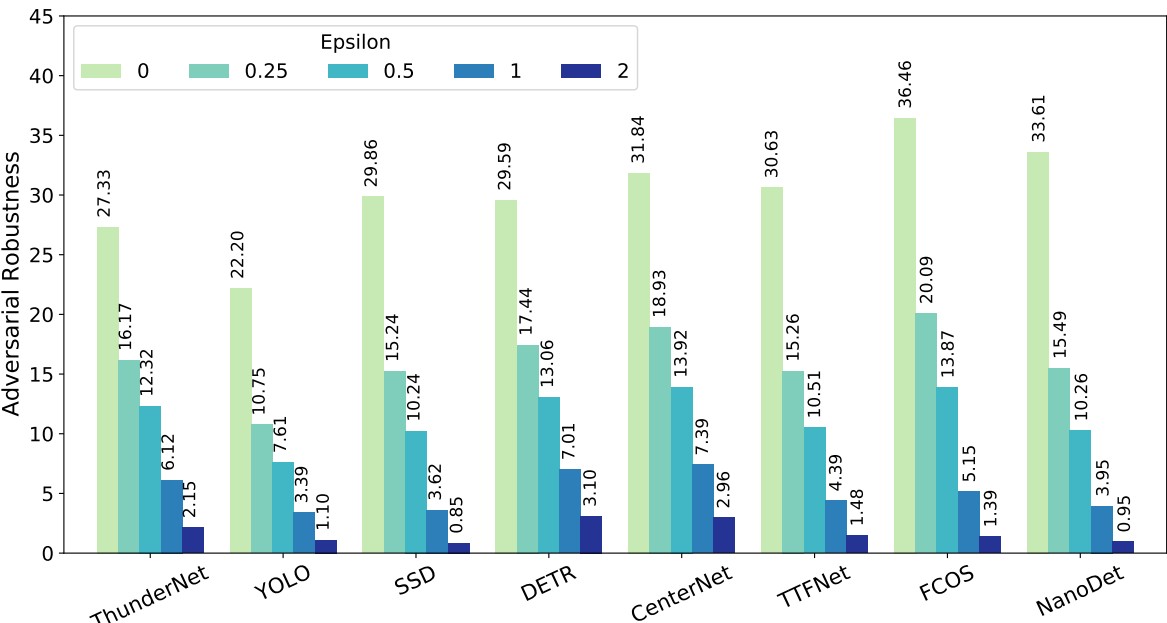

Figure 19: Adversarial robustness of all detection networks (with HarDNet-68 backbone) trained on COCO, against PGD attacks of varying strengths (Epsilon). Epsilon=0 represents the original natural accuracy.

keypoint-based detectors are relatively more robust to the natural corruptions than the other detectors, as seen from the upper clusters in all the graphs.

**Detailed Analysis:**

To provide a more detailed analysis, we show results on all the fifteen different corruptions for each network in Figure 22. In each of the heatmaps, we calculate the mean Corruption Accuracy (mCA) by averaging over all the corruptions. All the detectors show a similar decline trend in performance across all the three noises (gaussian, shot and impulse noise). *FCOS* and *TTFNet* have the least decline and are relatively more robust to noisy corruptions compared to others. Amongst the blur corruptions, the decline is more steady for defocus and motion blur while for the glass blur, the accuracy declines gradually initially but dips severely after severity level 3. On the zoom blur, the dip in performance of all detectors starts right from severity level 1. All the detectors are robust to varying brightness and fog corruptions than on frost and snow. The worst performance is seen in snowy conditions and the trend is similar across all. Among the digital effects, networks are more robust to elastic transformation and JPEG compression compared to pixelation and contrast. All the models are less robust to contrast changes with *YOLO* being the least robust. The visualizations of the predictions of each network on the different types of corruptions are presented in Figure 24 in Appendix.

## 12   Adversarial Robustness

Several works have shown the vulnerability of deep neural networks to adversarial attacks. Adversarial perturbations are imperceptible noise, that when added to the data, appear imperceptible to human eye, but can lead the networks to make wrong predictions. In safety-critical applications such as AD, robustness is of more importance to prevent networks from making untimely decisions. Hence, robustness against adversarial attacks is a critical metric for object detection. However, it is not prominently present in the literature. Here, we evaluate the robustness of all the eight detector networks against adversarial attack.

Table 9: Results of detection models with HarDNet-68 backbone trained on BDD and tested on BDD (in-distribution test) and CityScape (out-of-distribution test) datasets. MAC count, number of parameters, inference energy consumption (kJ), mAP (@IoU: 0.7), and inference speed (FPS) are reported. The best performance for each metric is highlighted.

| **Head** | MAC | #Params | **BDD** | | | **Cityscapes** | | |
|---|---|---|---|---|---|---|---|---|
| | (G) | (M) | mAP | FPS | kJ | mAP | FPS | kJ |
| ThunderNet | 22.96 | 17.68 | 21.51 | 45.35 | 47.94 | 18.69 | 44.26 | 4.44 |
| YOLO | 30.17 | 47.34 | 6.38 | 42.01 | 50.53 | 4.47 | 36.46 | 4.44 |
| SSD | 24.27 | 23.46 | 26.15 | 10.69 | 29.97 | 20.07 | 11.58 | 3.16 |
| DETR | 25.77 | 38.40 | 13.80 | 40.10 | 60.13 | 11.29 | 36.26 | 5.45 |
| CenterNet | 36.58 | 32.76 | 24.19 | 73.18 | 44.23 | 19.22 | 65.62 | 4.86 |
| TTFNet | 69.57 | 36.14 | 23.05 | 58.56 | 52.94 | 18.74 | 50.09 | 5.69 |
| FCOS | 126.99 | 41.11 | 27.89 | 30.63 | 93.23 | 23.05 | 28.08 | 9.16 |
| NanoDet | 22.60 | 16.81 | 30.09 | 55.51 | 47.12 | 22.88 | 48.55 | 4.69 |

We employ gradient based attacks which utilize the gradient information of the network to generate the perturbation. Projected Gradient Descent (PGD) (Madry et al., 2017), a common untargeted attack, maximizes the training loss to generate an adversarial perturbation, which is clamped within an epsilon bound. We use both classification loss and regression loss as the objective for the PGD attack. We perform the PGD attack at varying attack strengths and report the resulting accuracy in Figure 19. The accuracy at Epsilon=0 refers to clean accuracy on the original test set. As the attack strength increases, the performance declines. *CenterNet* and *DETR* exhibit consistent and better robustness compared to the other detectors. *FCOS* has the highest natural accuracy and shows good resistance to the very weak attacks but the performance sharply decline for the higher perturbations. *TTFNet* and *ThunderNet* are the next best performers. *YOLO*, *NanoDet*, and *SSD* occupy the next spectrum.

# 13    Case Study: Autonomous Driving

Real-time object detection is highly pertinent in the autonomous driving (AD) domain and the network needs to learn various objects such as pedestrians, vehicles, and road signs present on city roads and highways. Most benchmarks for detection networks are provided on VOC and COCO datasets, which consist of mostly household objects. Results on these datasets will not suffice in gauging the capability of the networks to perform in an AD setting. Hence, we conduct a realistic case study for AD using the BDD dataset (Yu et al., 2018) which is one of the largest and most diverse datasets in this domain. First, we show the performance of all the networks on this complex dataset. Then, we address the Out-of-Distribution (OOD) generalization by using the models trained on BDD and testing on a different dataset (i.e. Cityscapes (Cordts et al., 2016)). Finally, we deploy all the models trained on BDD on the embedded devices and report the numbers to showcase the real-time application capability of each network. As shown in Section 9.5, the accuracy gain due to DCN is not significant compared to the decline in speed. Hence in this section, we consider all the networks uniformly without DCN layers.

## 13.1    Generalization on IID Data

In Table 9, we present the results obtained on the BDD validation set. Similar to Zhao et al. (2018a), we computed the accuracy (mAP) of our models at $IoU = 0.7$. *NanoDet* exhibits the best accuracy. *FCOS* is the next most accurate but falters in speed and also has the highest energy consumption. *CenterNet* is the fastest with marginally lower accuracy. *SSD* consumes the least energy and *YOLO* has the lowest accuracy. Interestingly, the bias of object positions in the BDD dataset results in generating lesser region proposals thus making ThunderNet faster. The qualitative results of each detector on the BDD in Appendix, Figure 25.

Table 10: Inference speed (FPS) of all TensorRT optimized detection networks with HarDNet-68 backbone on BDD dataset deployed on three devices at FP32, FP16, and INT8 precision (note that INT8 is not supported on TX2). The best performances are highlighted.

| Head | RTX 2080Ti | | | Xavier | | | TX2 | |
|------|------|------|------|------|------|------|------|------|
| | FP32 | FP16 | INT8 | FP32 | FP16 | INT8 | FP32 | FP16 |
| ThunderNet | 151 | 212 | 225 | 15 | 30 | 36 | 10 | 16 |
| YOLO | 174 | 244 | 280 | 15 | 33 | 45 | 9 | 17 |
| SSD | 154 | 212 | 225 | 15 | 30 | 36 | 10 | 16 |
| DETR | 138 | 184 | 184 | 14 | 29 | 30 | 8 | 13 |
| CenterNet | 141 | 229 | 228 | 11 | 29 | 29 | 7 | 13 |
| TTFNET | 101 | 193 | 206 | 7 | 19 | 19 | 4 | 8 |
| FCOS | 53 | 116 | 133 | 4 | 10 | 13 | 2 | 4 |
| NanoDet | 153 | 210 | 213 | 15 | 27 | 33 | 9 | 14 |

## 13.2 Generalization on OOD Data

Generalization to distribution shift is one of the main challenges in AD scenarios. The networks, when deployed in real-life applications, need to adapt to unseen data and perform consistently. However, a majority of deep learning benchmarks are shown on the test set, which has the same distribution as the training data (Geirhos et al., 2020). Therefore, to test the robustness of networks to distribution shift, we test the BDD trained models on Cityscapes data. We extract the ground-truth bounding boxes from the instance segmentation annotations of the Cityscapes dataset. In Table 9, we observe that *FCOS* has the best accuracy with *NanoDet* coming a close second. *CenterNet* is the fastest network and *SSD* is the most energy efficient in both the sets. In general, keypoint-based detectors are good candidates for generalization across challenging AD datasets.

## 13.3 Performance on Embedded Devices

AD applications have power and resource constraints as the networks are deployed on the on-board edge devices. The real-time performance of detection networks on low-power devices is paramount to their efficacy. For deployment, we use the TensorRT library (ten) to convert the networks into optimized high-performance inference engines. TensorRT is NVIDIA's parallel programming model that can optimize neural networks for deployment on embedded or automotive product platforms. These engines are then tested on three different ranges of GPUs by NVIDIA: (1) 2080Ti, a commonly used desktop GPU (2) Jetson-Xavier, a powerful mobile GPU, and (3) Jetson-TX2, a low-power mobile GPU.

Table 10 shows the inference speed of all the eight detectors for three precision modes, i.e. FP32, FP16, and INT8. The performance trend might not be the same as seen earlier, as it depends on the optimization of different layers by TensorRT. The optimization fuses subsequent layers and makes the computation parallel. The anchor-based detectors *ThunderNet*, *YOLO*, and *SSD* have relatively simple architectures and achieve the highest gain in speed after optimization. *YOLO* being the least complex gets optimized the most and is the fastest across all the platforms. However, all the keypoint-based detectors obtain the least gain in speed from optimization. *DETR* lies in the middle spectrum and since the transformer architecture is relatively new it is not as optimized by the TensorRT engine as other convolution layers.

This unique case study reveals that the performance trend seen on one device does not necessarily translate to other hardware. This benchmark proves useful when choosing models to deploy on edge devices for real-time AD applications.

## 14 Case Study: Healthcare

The recent advancements in deep learning have enabled AI models to assist surgeons and radiologists in diagnosing and treating life-threatening diseases. Manual detection requires expertise, takes time, and can

Table 11: Results of detection networks with HarDNet-68 backbone trained and tested on Kvasir-SEG. MAC count, number of parameters, inference energy consumption, mAR and mAP (@IoU: [0.5-0.95]), and inference speed (FPS) are reported. The best performance for each metric is highlighted.

| **Head** | MAC (G) | #Params (M) | Inf Energy (kJ) | Inf Speed (FPS) | mAR | mAP |
|---|---|---|---|---|---|---|
| ThunderNet | 22.92 | 17.63 | $4.34_{\pm 0.02}$ | $51.79_{\pm 2.01}$ | $89.53_{\pm 0.25}$ | $67.49_{\pm 1.14}$ |
| YOLO | 30.16 | 47.29 | $5.05_{\pm 0.02}$ | $53.16_{\pm 0.95}$ | $94.10_{\pm 1.02}$ | $60.25_{\pm 1.34}$ |
| SSD | 23.02 | 21.66 | $4.02_{\pm 0.02}$ | $68.70_{\pm 1.27}$ | $90.42_{\pm 0.25}$ | $66.98_{\pm 0.82}$ |
| DETR | 25.77 | 38.39 | $5.07_{\pm 0.05}$ | $37.85_{\pm 1.11}$ | $89.97_{\pm 0.68}$ | $63.70_{\pm 1.04}$ |
| CenterNet | 33.38 | 33.19 | $4.44_{\pm 0.04}$ | $55.99_{\pm 1.02}$ | $92.33_{\pm 0.68}$ | $69.30_{\pm 0.93}$ |
| TTFNet | 65.16 | 36.62 | $5.31_{\pm 0.06}$ | $43.58_{\pm 0.47}$ | $91.30_{\pm 0.51}$ | $65.58_{\pm 1.30}$ |
| FCOS | 126.78 | 41.07 | $6.76_{\pm 0.03}$ | $26.76_{\pm 0.54}$ | $94.84_{\pm 1.28}$ | $71.42_{\pm 0.79}$ |
| NanoDet | 22.60 | 16.80 | $4.07_{\pm 0.03}$ | $65.02_{\pm 1.20}$ | $91.59_{\pm 0.00}$ | $70.44_{\pm 1.08}$ |

also be subjected to human errors. AI-based detection solutions aid in reducing cost, and resources, and can provide an accurate tool for detection in medical imaging.

One of the applications is to use DNNs to detect polyps in medical images. Colon and rectum (colorectal) cancer is commonly caused by the polyps found on the inner lining of the colon or rectum. Detecting those polyps and treating them at a very early stage is vital for cancer treatment. The medical images have a drastically different distribution than the standard datasets such as COCO and VOC. Hence, standard benchmarks may not be able to provide vital information about which model to choose for this application. Also, different metrics are more relevant depending on the application. While standard benchmarks focus on the precision metric for accuracy, in the healthcare industry, where even one false negative can cause more damage than a false positive result, recall is more important.

To address this new data distribution and metrics, we conduct a case study specifically for medical images by evaluating the detectors on the Kvasir-SEG dataset. Table 11 presents the results obtained on the test split of Kvasir-SEG. The recall is more relevant to this application and we, therefore, report mean average recall (mAR) along with mAP. Certain networks like *YOLO* may not have the highest precision but fare well w.r.t recall. *FCOS* has the highest recall and precision which make it an ideal candidate for such test cases. In terms of speed, *SSD* is the fastest, while *Nanodet* comes second. The qualitative results of each detector are presented in Figure 26 in Appendix.

## 15 Discussion

We provide a comprehensive study of a combination of feature extractors and detectors (ranging from two-stage, single-stage, anchor-based, keypoint-based, and transformer-based architectures) under a uniform experimental setup across different datasets. We report an extensive set of results including accuracy, speed, resources, and energy consumption, also robustness and calibration analyses. We evaluate the robustness of detectors against both natural adversarial corruptions. Additionally, detailed insights are highlighted to get a complete understanding of the effect of different variables on the final result. Different variables such as the effect of the backbone architecture, image size, object size, confidence threshold, and specific architecture layers are decoupled and studied. We also contribute two unique case studies on two diverse industries: autonomous driving and healthcare. We, further, optimize and benchmark the networks on embedded hardware to check the feasibility of the networks to be deployed on edge devices.

The combination of results in Section 8, suggests that keypoint-based detectors tend to generalize well across multiple datasets, as anchor box optimization is no longer required. *NanoDet* fares well in terms of both accuracy and speed while also being resource-friendly. *CenterNet* is the second fastest and also lies in the good spectrum on all other metrics and *TTFnet* lies in the middle spectrum. *FCOS* has the highest accuracy but falters in other metrics, while *DETR*, which is the Transformer-based detector, lies in the middle spectrum. Among the backbones, modern networks designed specifically for low-memory traffic, such as

*HarDNet*, provide the best balance between accuracy, inference speed, and energy consumption. All detectors underperform when detecting small objects, with *FCOS* performing relatively better. Varying anchors affect the performance in a non-deterministic way, thus rendering them difficult to generalize. We report the accuracy-speed-resource requirement trade-off that should be taken into consideration when switching to higher image sizes or using DCN layers. On robustness against natural corruptions, the performance of all the networks deteriorates on all the fifteen corruptions, and it declines faster as the severity increases. In general, the keypoint-based detectors are relatively more robust to natural corruptions than the other detectors. *FCOS* is the most robust while *YOLO* is the least robust. *FCOS* and *TTFNet* are relatively more robust to noisy and blurry corruptions, but all detectors fail in snowy conditions. *CenterNet* proves to be the most robust against adversarial perturbations, while textit FCOS and *DETR* are also quite resistant to these attacks. In reliability analysis, *SSD* is relatively well calibrated, while keypoint-based detectors are more prudent in their predictions, thus rendering them useful in safety-critical applications. *ThunderNet* and *DETR* lean towards being more overconfident. Thus, the analysis of transformer-based detectors in these various settings will offer more insight into the capabilities and pitfalls of this new architectural paradigm. Case studies on AD and healthcare cover two important domains with different requirements. The AD case study reports the performance on a more relevant dataset related to driving scenarios and also reports the OOD generalization performance. The deployment on three different GPUs (desktop and embedded) shows that the performance on embedded hardware shows a different trend from the desktop GPUs. Anchor-based detectors, such as *SSD* and *YOLO* are better optimized owing to the simple architectures and run faster on edge devices. This study helps compare and choose a network based on the hardware capability of an application. The healthcare case study highlights the importance of the recall metric (compared to just precision results), and we observe that the most accurate network is not necessarily the best performer w.r.t. the recall metric. These case studies cover two diverse domains that have different requirements and offer a unique perspective on looking beyond the standard benchmarks and gauging the capability of the detectors on diverse datasets that are more relevant and applicable in real-time applications.

**Broader Impact Statement**

We provide a holistic and comprehensive analysis of deep learning-based real-time object detection networks across multiple datasets and domains in a standard pipeline. We showcase all the high-level generic results while also zooming in on encapsulated detailed intuitions. Our extensive analyses also provide insights into the capabilities and pitfalls of new architectural paradigms (transformers vs CNNs). Different applications have different criteria, and our study can act as a guideline for the industrial community to gauge the different trade-offs while choosing detectors for the respective application. And since new detection networks are being introduced frequently, we also hope to inspire the research community to use this study as a precept for new designs. This study highlights the importance of a standardized, transparent and fair pipeline and also emphasizes the need to shift the focus from nominal improvements to a more broad perspective. We hope this will help pave a new way for future research.

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

## A    Appendix

### A.1    Overall Performance Graph

Figure 1 shows the overall performances of all detectors across varying metrics. The results are from detectors trained on COCO dataset with image resolution 512 and HarDNet-68 backbone. Each vertex corresponds to a metric and the eight different colors represent different detection heads. We report eight metrics: accuracy (in terms of mAP), natural robustness (in terms of mAP), speed (in terms of FPS), number of parameters, MAC count, energy consumption (in terms of kJ), calibration error (that measures reliability), and adversarial robustness (in terms of mAP). For procuring one natural robustness value per detector, the accuracy of each detector (with HarDNet-68 backbone) is averaged over all the (fifteen) corruptions for all five levels of severity (Figure 22). For adversarial robustness, the robustness accuracy is averaged for all the four attack strengths (Epsilon: $[0.25, 0.5, 1, 2]$; Figure 19). The calibration errors are the D-ECE scores from Table 8. The rest of the metric values are obtained from the Table 4.

Considering an ideal application, a well performing detector is expected to have the highest accuracy, highest (natural/adversarial) robustness, highest speed, lowest number of parameters, lowest MAC count, lowest energy, and low calibration errors. Hence, for accuracy, robustness and speed metrics, naturally the higher values are better. But for the others, lower value is better. So, we use the inverse values for these metrics: number of parameters, MAC count, energy consumption, and calibration error (hence, represented with a superscript "$-1$" in the plot). We then normalize each metric between 0 and 1:

$$\hat{z}_i = \frac{z_i - min(z)}{max(z) - min(z)} \tag{19}$$

Therefore, the ideal network should occupy the whole octagon, i.e. have value of 1 for all metrics.

### A.2    Details of Implementations

Here, we briefly discuss the implementation challenges and details.

The official code of ThunderNet is not released. ShuffleNet-v2 was introduced in the paper but the ImageNet pretrained weights for this new backbone was not available and had to be trained, to reproduce the results and moreover, not all the hyperparameters are included in the paper which made this implementation very hard and challenging. Recently, some unofficial repository shared the TensorFlow and Pytorch implementation of ThunderNet.

YOLOv2 is implemented using DarkNet, which is an open source framework implemented in C and CUDA. But, this is in a different structure compared to other repositories and hence was challenging to use. There were many other unofficial repositories in PyTorch but none had completely reproduced the results, and had many open issues.

SSD has multiple good repositories and the hyperparameters were tabulated well enough to reproduce the result.

CenterNet and TTFNet use DCN layers and the original repositories had built these layers for older version of Pytorch. These needed to be re-built for new upgraded versions and were not easily available. TTFNet, designed to train on fewer epochs, needed some tuning to find the best epoch for a particular dataset. The settings given in the paper did not work for all datasets. TTFNet has faster training time but is a bit challenging to find the converging parameters.

NanoDet albeit having a good repository, does not have a publication. Using the repository was easy to reproduce the results but some concepts and loss functions were derived from multiple different works, and a paper publication would have enhanced the learning experience.

Both FCOS and DETR had a good repository and were easy to reproduce.

On the other hand, for TensorRT conversion, we need to follow two steps: (i) exporting the model to ONNX format, and (ii) converting the model from ONNX to TensorRT Engines. The complexity of these two

Table 12: The detailed setting of the experiments for each network. The initial learning rate (LR) is decayed by a factor of 0.1 and we use a common *batch_size* of 32 and *confidence_threshold* of 0.01 for all the experiments. BG stands for the background class and Opt stands for the optimizer.

| Head | BG | Opt. | # Iters: VOC/COCO/BDD/Med | LR: VOC/COCO/BDD/Med | LR Steps (% of #Iters)[1] | Warmup Iters | Weight Decay |
|---|---|---|---|---|---|---|---|
| ThunderNet | ✓ | SGD | 120K/400K/240K/50K | 0.005/0.005/0.005/0.005 | (70, 90) | 500 | 0.0001 |
| YOLO | ✗ | SGD | 120K/400K/240K/50K | 0.0001/0.0001/0.0001/0.0001 | (70, 90)[2] | 500 | 0.0005 |
| SSD | ✓ | SGD | 120K/400K/240K/50K | 0.001/0.001/0.005/0.001 | (70, 90) | 500 | 0.0005 |
| DETR | ✗ | AdamW | 120K/400K/240K/50K | 0.0001/0.0001/0.0001/0.0001 | 70 | 0 | 0.0001 |
| CenterNet | ✗ | SGD | 120K/400K/240K/50K | 0.001/0.001/0.005/0.001 | (70, 90) | 500 | 0.0001 |
| TTFNet | ✗ | SGD | 30K/120K[3]/72K/50K | 0.001/0.016/0.005/0.016 | (70, 90) | 500 | 0.0001[4] |
| FCOS | ✓ | SGD | 120K/400K/240K/50K | 0.001/0.001/0.005/0.001 | (70, 90) | 500 | 0.0005 |
| NanoDet | ✗ | SGD | 120K/400K/240K/50K | 0.001/0.001/0.005/0.001 | (70, 90) | 500 | 0.0005 |

steps is determined by the number of custom layers present in the model and their support availability in TensorRT Open Source Software (OSS). SSD, YOLO, FCOS, and NanoDet have only one custom layer at the end, i.e. the NMS layer, for which support is available in TRT-OSS. Hence, with some additional work, these models can be converted into TensorRT inference engines. CenterNet, TTFNet, and DETR do not have any custom layers, which makes them easier to convert without any additional work. The ThunderNet contains additional custom layers, such as ROI proposal and ROI align. Since these layers are not exactly supported in TRT-OSS, an equivalent layer has to be selected while exporting to ONNX which made the conversion very difficult and challenging. Note that apart from custom layers, functions such as "upsample" and "gather" are version sensitive in ONNX and TensorRT.

## A.3   Resource Analysis

The MAC count is a measure of the multiply-accumulation operations in neural networks. However, it has been brought to our attention that the standard flop counter libraries available online may not accurately account for MAC in transformer-based architectures that utilize multi-head attention modules. These libraries are typically designed for use with convolutional neural networks and do not consider transformer-based architectures.

To address this issue, we evaluated various flop counter libraries and ultimately chose to use the fvcore(v0.1.5) library [5] to calculate the flops of our transformer-based backbone (DEIT-T) and detector (DETR). This library was selected because it is capable of accurately accounting for MAC in transformer-based architectures, in addition to the common layers used in neural networks. We also found that the MAC counts for CNN-based architectures using the fvcore library were the same as those obtained using a previously used library.

## A.4   Reliability Analysis on OOD Data

Calibration for the IID COCO dataset is provided in Section 10. Here, we report the calibration results for the OOD version by testing the networks on the Corrupted COCO dataset in Section 7.1. COCO validation set consists of 5000 images and we apply a random corruption out of 15 corruptions to each of the sample and evaluate the calibration on this set. Reliability diagrams are provided in Figure 21 where the diagonal represents the perfect calibration and the green shades represent the gap in calibration.

---

[1]With the exception for VOC dataset where 67% of #Iter used for DETR and (67%, 83%) for all the other detectors.
[2]YOLO starts with a lower learning rate of $1e-4$ but it is slowly increased to $1e-2$ in the first few epochs.
[3]TTFNet-EfficientNet-B0 needs twice these epochs to converge
[4]With the exception for MED dataset where weight decay of 0.0005 is used.
[5]https://github.com/facebookresearch/fvcore

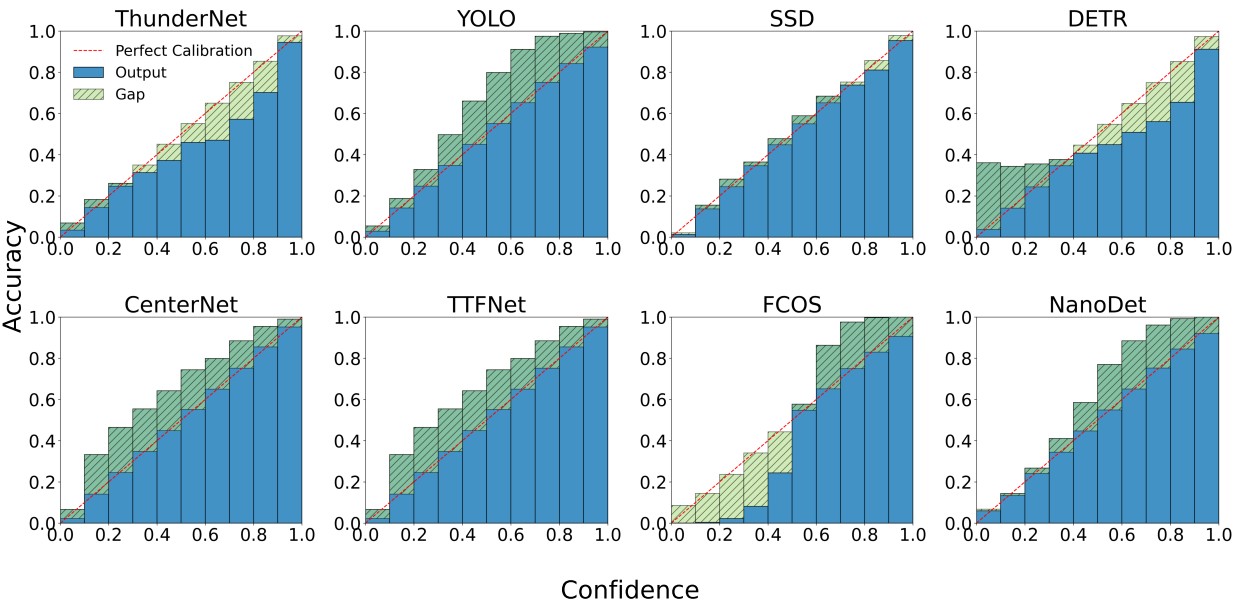

Figure 20: Reliability diagrams on VOC dataset for all detection heads with HarDNet-68 backbone. Green boxes indicate the error compared to perfect calibration (darker shade of green indicates the underconfident predictions whereas the lighter shade indicates the overconfident predictions). The better the calibration is the more reliable the predictions are.

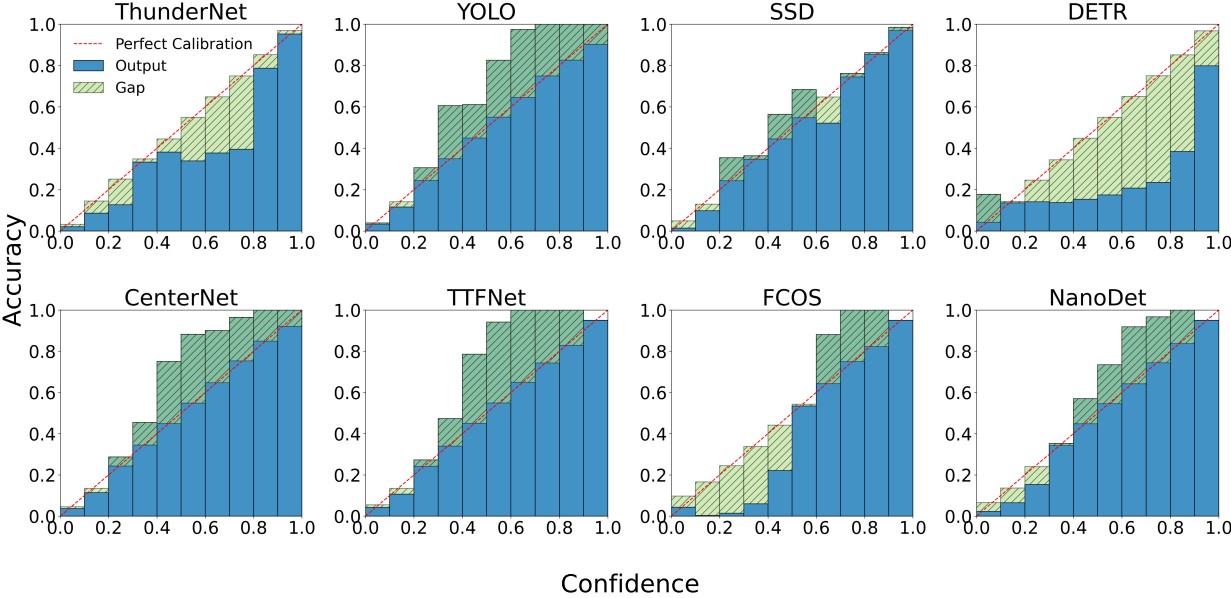

Figure 21: Reliability diagrams (based on classification prediction) of detection networks with HarDNet-68 backbone trained on COCO dataset and tested on Corrupted COCO dataset. Green (striped) boxes indicate the error compared to perfect calibration - darker shade of green indicates the underconfident predictions whereas the lighter shade indicates the overconfident predictions. The better the calibration, the more reliable the detector is. *SSD* is relatively well calibrated regarding an OOD data as well. Overall, in line with the IID reliability analyses, the single-stage detectors are more underconfident while the two-stage *ThunderNet* and *DETR* are very overconfident.

**ThunderNet**

| | | Noise | | | Blur | | | | Weather | | | | Digital | | | |
|---|---|---|---|---|---|---|---|---|---|---|---|---|---|---|---|---|
| Severity | 0 | 27.3 | 27.3 | 27.3 | 27.3 | 27.3 | 27.3 | 27.3 | 27.3 | 27.3 | 27.3 | 27.3 | 27.3 | 27.3 | 27.3 | 27.3 |
| | 1 | 22.4 | 23.5 | 21.5 | 23.5 | 22.2 | 22.0 | 22.8 | 11.6 | 26.5 | 24.6 | 22.3 | 20.5 | 24.8 | 22.0 | 24.1 | 24.6 |
| | 2 | 20.0 | 21.1 | 18.5 | 20.8 | 20.1 | 18.5 | 19.7 | 8.1 | 25.9 | 23.9 | 18.2 | 15.3 | 23.6 | 19.5 | 22.4 | 24.0 |
| | 3 | 16.7 | 16.8 | 16.3 | 16.9 | 16.0 | 9.6 | 15.1 | 5.8 | 25.1 | 23.0 | 15.1 | 14.7 | 21.2 | 16.3 | 20.9 | 18.5 |
| | 4 | 13.1 | 11.5 | 10.5 | 10.6 | 12.6 | 7.8 | 10.2 | 4.3 | 23.9 | 22.5 | 14.5 | 11.2 | 14.4 | 14.1 | 16.3 | 12.0 |
| | 5 | 9.7 | 5.4 | 5.3 | 6.7 | 9.6 | 5.9 | 7.5 | 3.2 | 22.3 | 20.1 | 12.9 | 10.5 | 6.4 | 11.2 | 10.6 | 7.8 |
| | | mCA | Gaussian | Impulse | Shot | Defocus | Glass | Motion | Zoom | Brightness | Fog | Frost | Snow | Contrast | Elastic | JPEG | Pixelate |

**YOLO**

| | | Noise | | | Blur | | | | Weather | | | | Digital | | | |
|---|---|---|---|---|---|---|---|---|---|---|---|---|---|---|---|---|
| Severity | 0 | 22.2 | 22.2 | 22.2 | 22.2 | 22.2 | 22.2 | 22.2 | 22.2 | 22.2 | 22.2 | 22.2 | 22.2 | 22.2 | 22.2 | 22.2 | 22.2 |
| | 1 | 18.5 | 19.0 | 17.1 | 19.1 | 18.5 | 19.4 | 19.1 | 9.7 | 21.5 | 20.1 | 18.8 | 16.4 | 19.9 | 19.1 | 19.7 | 20.4 |
| | 2 | 16.4 | 16.9 | 15.0 | 16.7 | 16.5 | 15.9 | 16.0 | 6.5 | 21.0 | 19.3 | 15.1 | 12.6 | 18.9 | 16.8 | 18.0 | 20.0 |
| | 3 | 13.6 | 13.8 | 13.3 | 13.8 | 12.7 | 7.1 | 11.6 | 4.6 | 20.4 | 18.6 | 12.3 | 11.3 | 16.4 | 13.9 | 16.9 | 17.4 |
| | 4 | 10.7 | 9.4 | 8.8 | 8.8 | 9.5 | 5.1 | 7.4 | 3.2 | 19.7 | 18.3 | 11.9 | 8.5 | 10.5 | 11.9 | 12.5 | 14.3 |
| | 5 | 7.9 | 4.5 | 4.6 | 5.5 | 6.7 | 3.3 | 5.1 | 2.2 | 18.5 | 16.7 | 10.2 | 9.0 | 3.8 | 9.3 | 7.9 | 11.6 |
| | | mCA | Gaussian | Impulse | Shot | Defocus | Glass | Motion | Zoom | Brightness | Fog | Frost | Snow | Contrast | Elastic | JPEG | Pixelate |

**SSD**

| | | Noise | | | Blur | | | | Weather | | | | Digital | | | |
|---|---|---|---|---|---|---|---|---|---|---|---|---|---|---|---|---|
| Severity | 0 | 29.9 | 29.9 | 29.9 | 29.9 | 29.9 | 29.9 | 29.9 | 29.9 | 29.9 | 29.9 | 29.9 | 29.9 | 29.9 | 29.9 | 29.9 | 29.9 |
| | 1 | 23.8 | 24.6 | 21.4 | 24.9 | 23.6 | 23.8 | 24.3 | 11.7 | 28.8 | 26.3 | 23.8 | 21.2 | 26.5 | 24.5 | 25.1 | 25.9 |
| | 2 | 20.7 | 21.4 | 18.3 | 21.2 | 21.3 | 19.5 | 20.5 | 7.9 | 27.9 | 25.5 | 19.1 | 15.0 | 25.0 | 21.6 | 22.1 | 24.7 |
| | 3 | 16.9 | 16.3 | 15.4 | 16.3 | 16.5 | 9.8 | 15.2 | 5.8 | 26.7 | 24.2 | 15.5 | 14.6 | 22.2 | 17.7 | 19.7 | 17.6 |
| | 4 | 12.7 | 10.0 | 8.9 | 9.0 | 12.3 | 7.8 | 10.0 | 4.2 | 25.1 | 24.0 | 15.1 | 10.6 | 14.8 | 15.2 | 13.3 | 10.6 |
| | 5 | 9.2 | 3.5 | 3.5 | 4.6 | 9.2 | 5.7 | 7.1 | 3.1 | 23.3 | 21.5 | 13.2 | 10.2 | 6.7 | 12.0 | 7.8 | 6.1 |
| | | mCA | Gaussian | Impulse | Shot | Defocus | Glass | Motion | Zoom | Brightness | Fog | Frost | Snow | Contrast | Elastic | JPEG | Pixelate |

**DETR**

| | | Noise | | | Blur | | | | Weather | | | | Digital | | | |
|---|---|---|---|---|---|---|---|---|---|---|---|---|---|---|---|---|
| Severity | 0 | 29.6 | 29.6 | 29.6 | 29.6 | 29.6 | 29.6 | 29.6 | 29.6 | 29.6 | 29.6 | 29.6 | 29.6 | 29.6 | 29.6 | 29.6 | 29.6 |
| | 1 | 24.6 | 25.4 | 22.7 | 25.8 | 25.4 | 25.2 | 26.0 | 12.6 | 28.9 | 26.9 | 24.2 | 21.2 | 27.3 | 23.7 | 26.6 | 26.7 |
| | 2 | 21.8 | 22.7 | 19.3 | 22.5 | 23.2 | 21.1 | 22.4 | 8.3 | 28.1 | 26.3 | 19.8 | 15.8 | 26.2 | 20.7 | 24.9 | 26.0 |
| | 3 | 18.1 | 17.8 | 16.4 | 17.9 | 18.7 | 10.3 | 17.2 | 5.9 | 27.3 | 25.1 | 16.1 | 14.6 | 23.8 | 16.7 | 23.6 | 20.9 |
| | 4 | 14.1 | 11.3 | 9.9 | 10.0 | 14.2 | 8.4 | 11.2 | 4.1 | 25.9 | 24.9 | 15.9 | 10.3 | 16.8 | 13.9 | 19.1 | 15.3 |
| | 5 | 10.3 | 4.6 | 4.5 | 5.6 | 10.3 | 5.9 | 8.1 | 2.9 | 24.2 | 22.2 | 13.6 | 10.9 | 7.7 | 10.9 | 13.2 | 9.9 |
| | | mCA | Gaussian | Impulse | Shot | Defocus | Glass | Motion | Zoom | Brightness | Fog | Frost | Snow | Contrast | Elastic | JPEG | Pixelate |

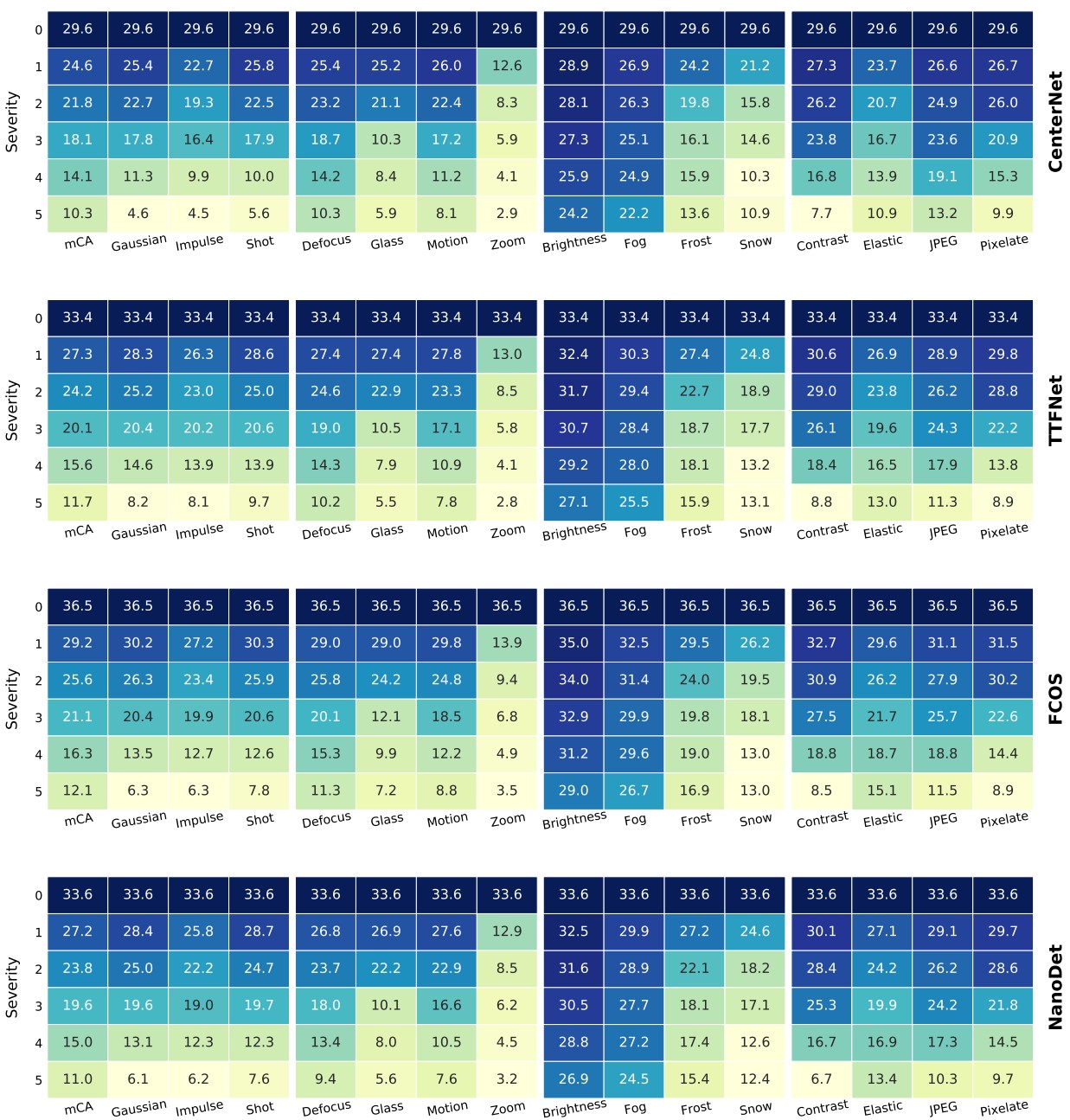

Figure 22: Robustness accuracy of all the eight detection heads with HarDNet-68 backbone when tested on Corrupted COCO dataset. The corruption has 5 levels of severity, with level 0 being the result on original COCO data.

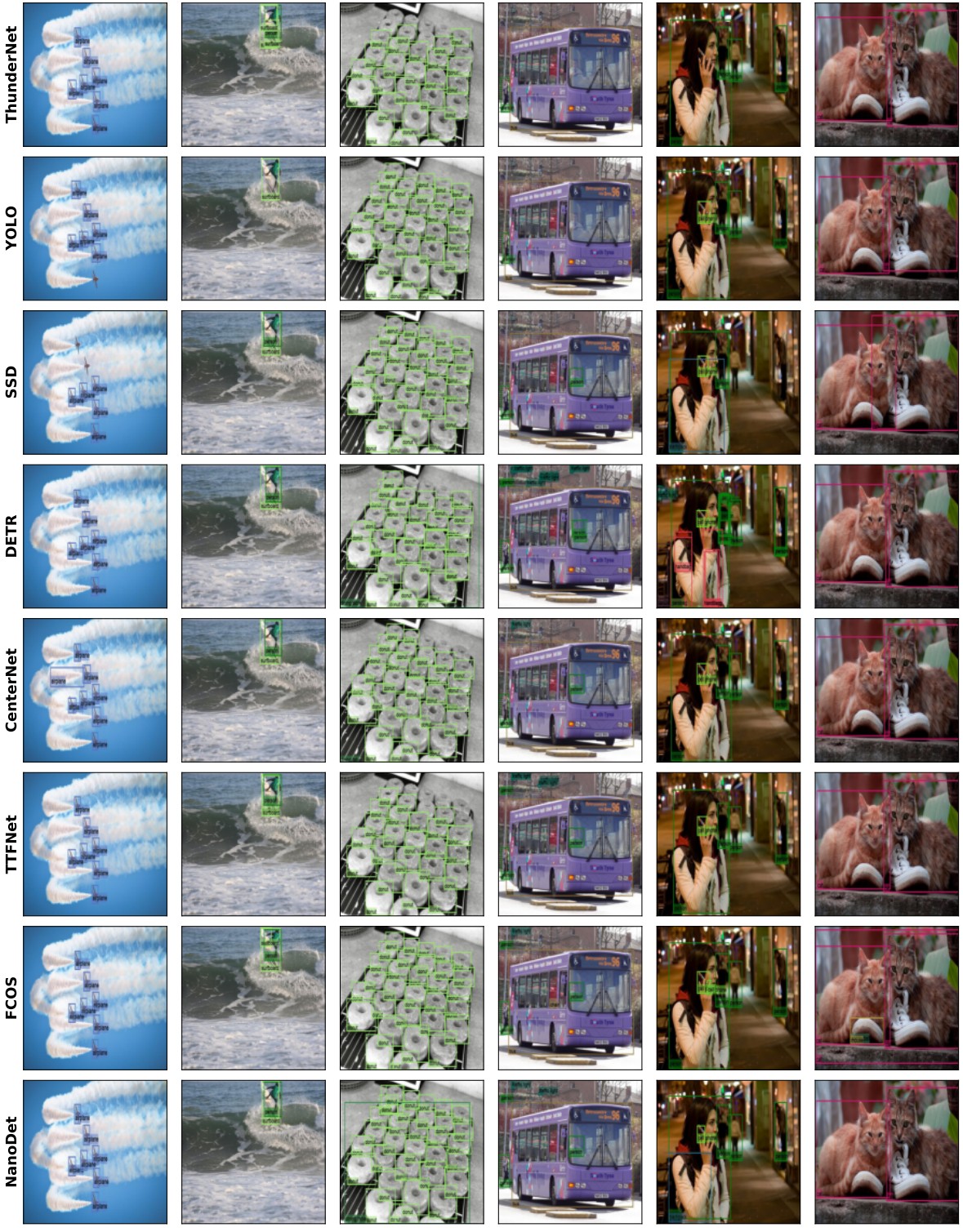

Figure 23: Visualization of predictions of all eight heads on COCO dataset.

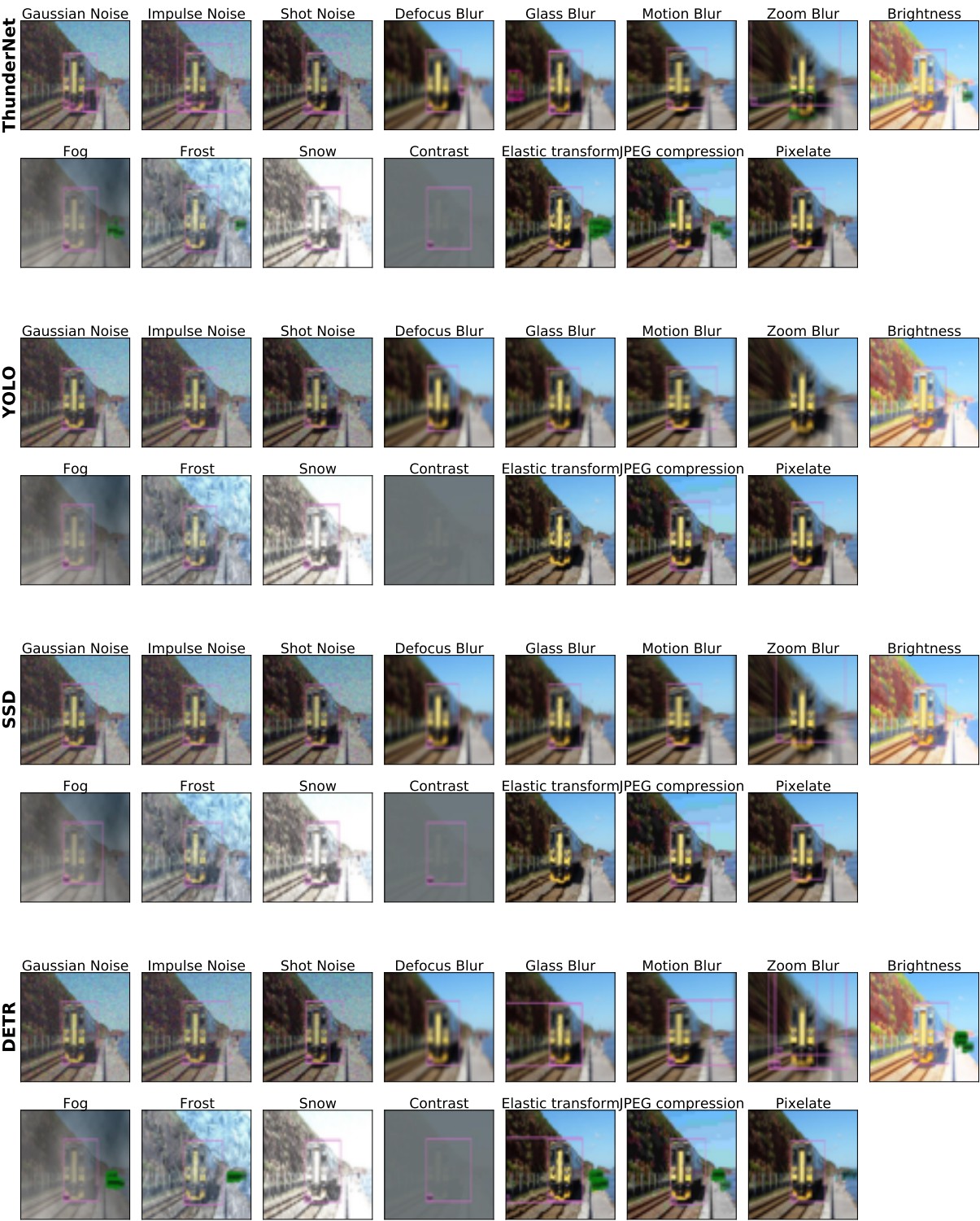

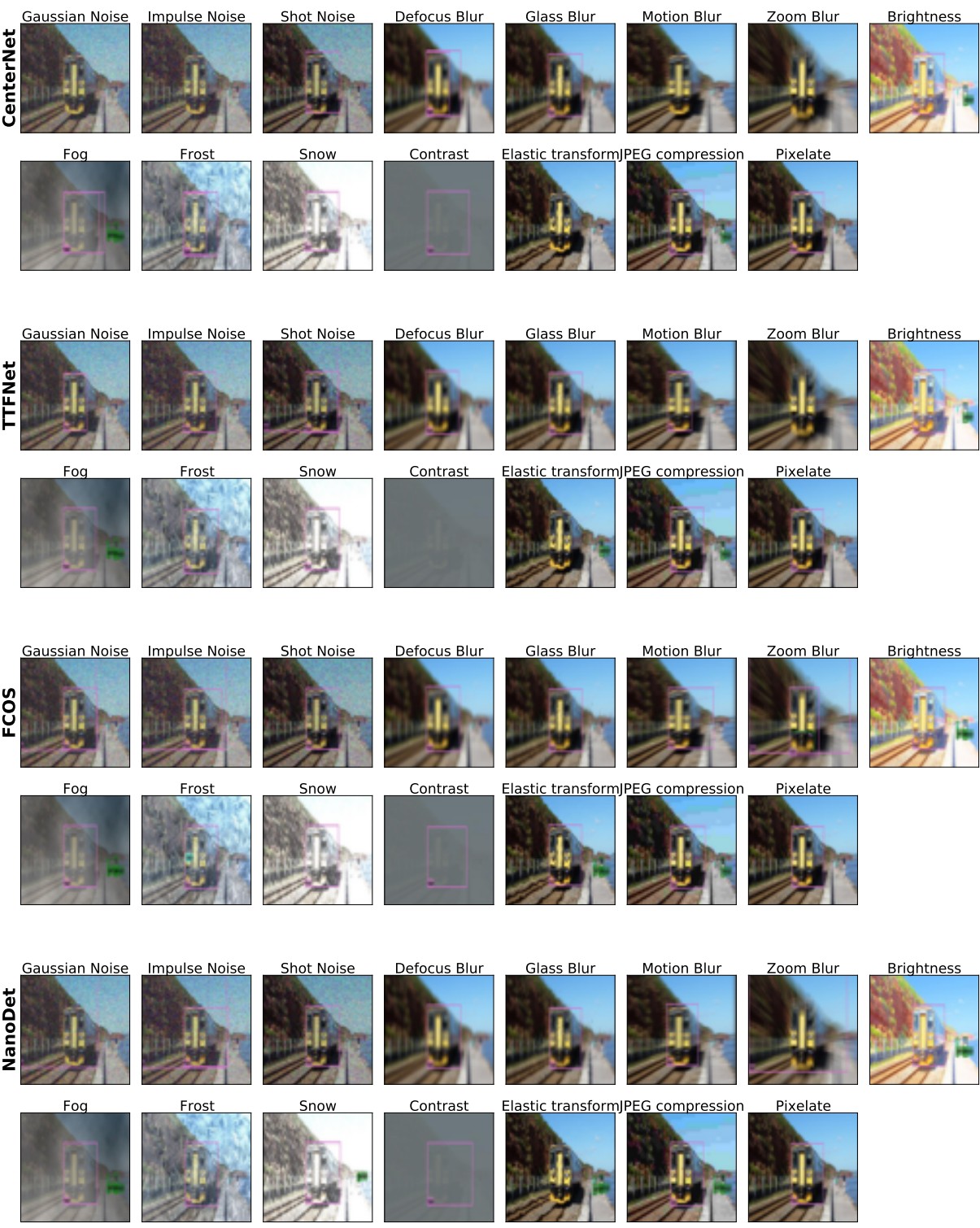

Figure 24: Robustness accuracy of the eight detection heads with HarDNet-68 backbone when tested on Corrupted COCO dataset. The corruption has 5 levels of severity, with level 0 being the result on original COCO data.

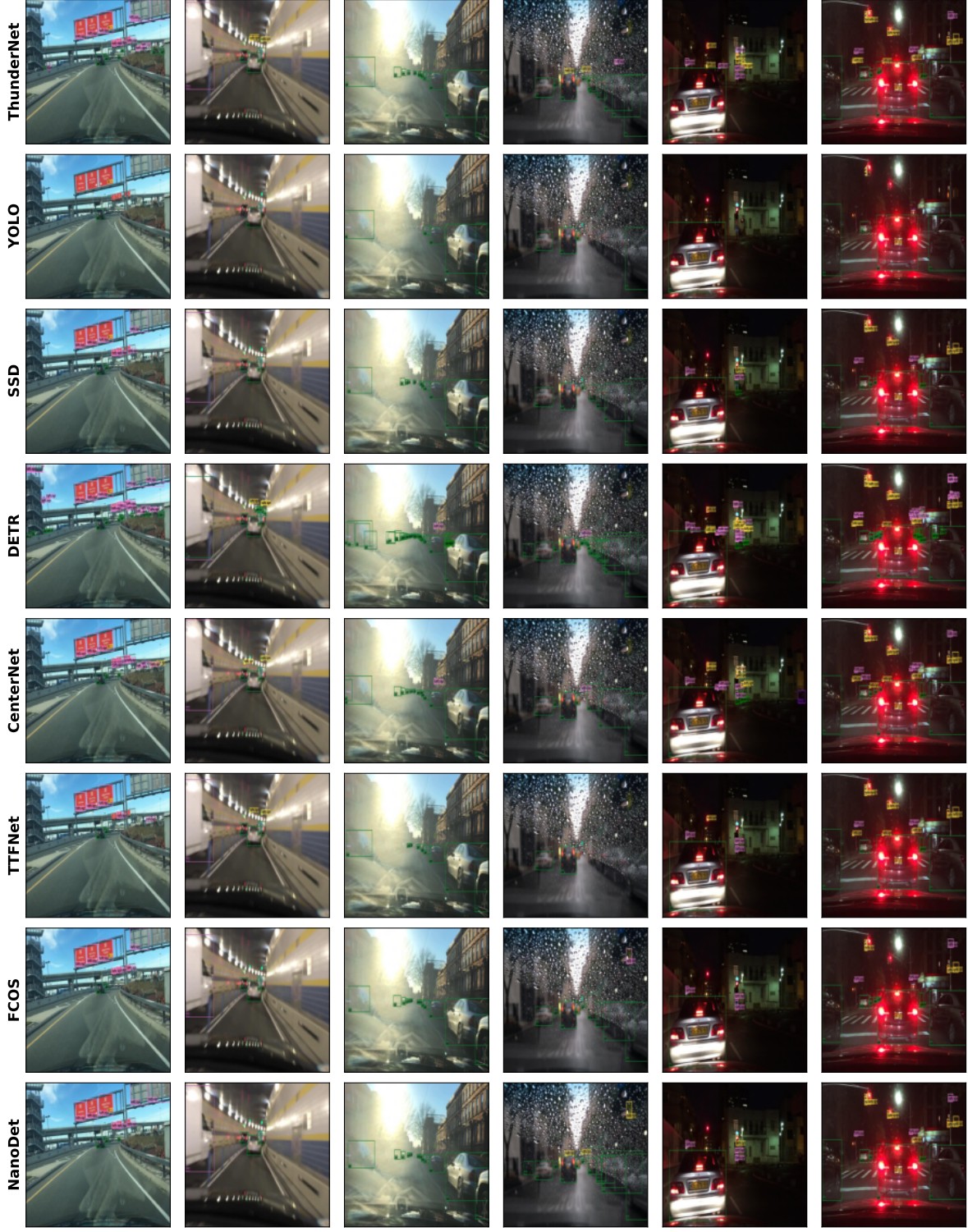

Figure 25: Visualization of predictions of all eight heads on BDD dataset.

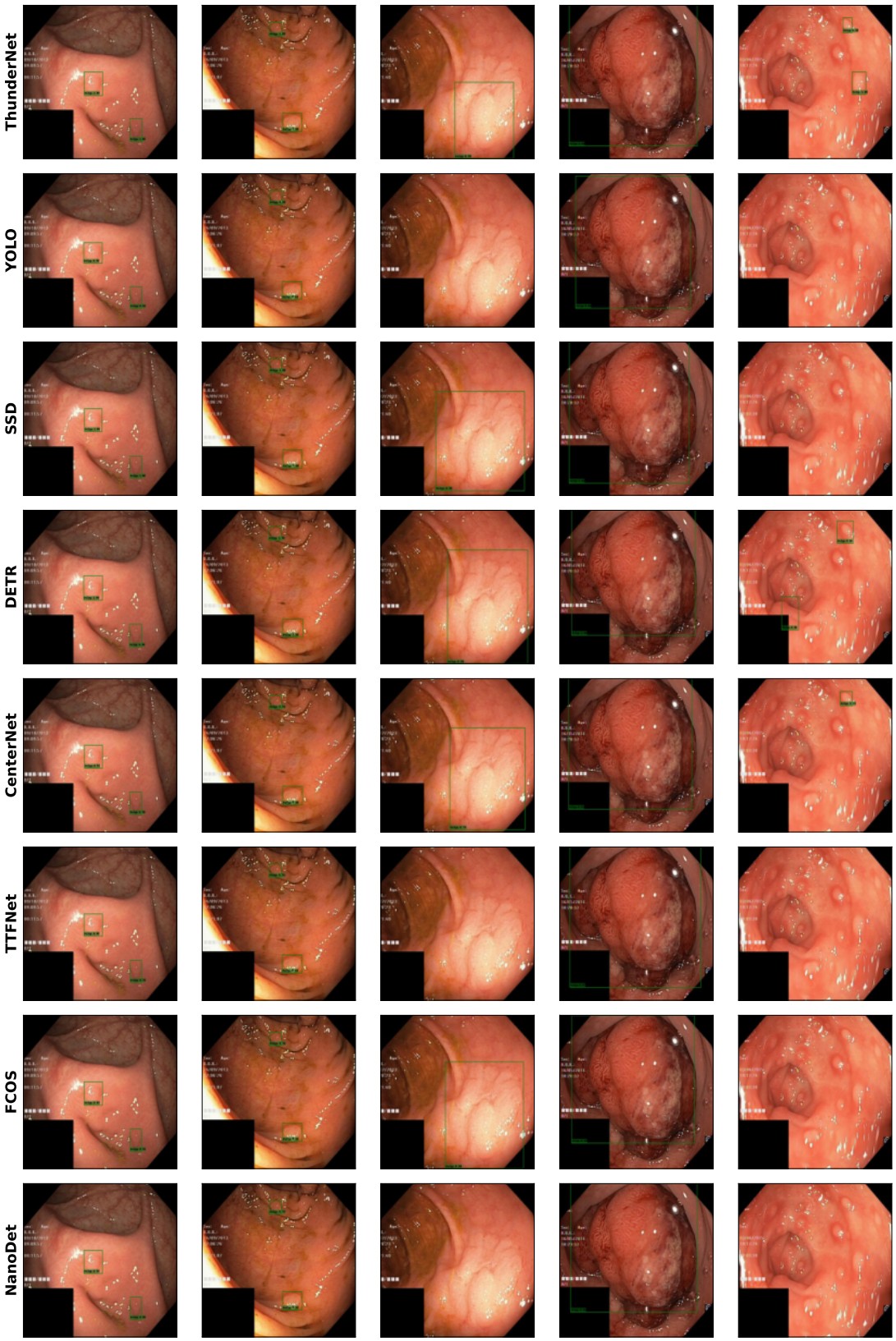

Figure 26: Visualization of predictions of all eight heads on Kvasir-SEG medical dataset.

