# OpenReview forum: "A Comprehensive Study of Real-Time Object Detection Networks Across Multiple Domains: A Survey"
_TMLR — Accepted by TMLR_

### Review · Reviewer_fya1 · 2022-06-16

**Summary Of Contributions:**

The paper is a very detailed survey on the study of real time Object Detection Networks. The paper details and evaluates different detection heads with different backbone, different dataset and different hardware. There is no methodological novelty but the proposed study deals with an interesting problem in many aspects with many experiments.

**Broader Impact Concerns:**

Nothing special to mention .

**Requested Changes:**

I would like to see a study on overfitting but this is not critical as the paper is already very exhaustive.

**Strengths And Weaknesses:**

Strengths:
- The paper is well written and easy to follow.
- The study is clear and very well detailed. The paper is very complete with a lot of evaluation with different architecture and different dataset. I think this can be of great value to the community.
- The applications considered are interesting

Weaknesses:
-  Overfitting evaluation: It would be interesting to do an overfitting study by testing the different approaches considered with several seeds to better understand the significance of the results. Since there are a lot of approaches and a lot of datasets, maybe do it with a not too expensive configuration just to get an idea of the overfitting.

---

> ### Author Response · Authors · 2022-07-10
> **Response to Reviewer fya1**
>
> Thank you for your feedback. We have added an overfitting study in the revised paper. We however chose the smaller medical imaging dataset due to time constraints. We trained all 8 detectors (with one backbone-HarDNet68) on the Kvasir-SEG dataset for two more seeds and added the mean and standard deviation over the three seeds for the relevant metrics in Table 11 (changes in blue).

---

### Review · Reviewer_CDPx · 2022-06-21

**Summary Of Contributions:**

The paper is a survey of real-time object detection networks that, in addition to the standard accuracy,  documents the energy consumption, robustness, calibration, and speed of a large number of popular and recent object detection networks.  The goal is to aid practitioners in choosing an object detection network suitable for their particular real-time application and to encourage researchers to evaluate future models across a more comprehensive set of metrics.

Contributions:

- This survey paper sets itself apart from prior surveys of object detection networks by focusing on a holistic set of metrics tailored to real-time applications and by considering a more modern set of networks, including newer anchor-free and transformer-based detectors.
- Network backbones and detection heads are evaluated in several combinations with a standardized plug-and-play pipeline across a large set of datasets.
- The authors test robustness of the networks to corruptions on the Corrupted COCO dataset, as well as OOD generalization in a self-driving application.
- They also evaluate calibration, as well as the impact of object size, image size, anchor box size, confidence thresholds, deformable convolution layers.
- Two case studies for health care and self-driving applications are considered.


**Requested Changes:**

- Give more explanation for how Figure 1 was created. Consider adding robustness to ood shifts as a vertex.
- Include an analysis of calibration on corrupted COCO shifts and/or the self driving OOD generalization dataset.
- Think more carefully about how to evaluate ease of use or adoption, code complexity, amount. of tuning needed, and simplicity. Include a more in depth discussion or evaluation if possible.

**Strengths And Weaknesses:**


Strengths:
- Overall, I found the survey comprehensive and well-executed. I think it will serve as a valuable resource for practitioners and researchers alike.
- The authors make several insightful comments about the field in general when motivating their work, e.g. “As newer detectors are being proposed frequently, without a standard evaluation framework, the trend is moving more towards a short-sighted direction that focuses only on nominal improvements.”
- The standardized evaluation protocol for a large combination of backbones and detector heads  is especially valuable in a field where minor implementation details across different papers make apples-to-apples comparisons difficult.

Weaknesses:
- Figure 1 is a nice overview of different networks, but leaves out robustness to distribution shifts as a key metric. It's also not clear how each metric/vertex was obtained - i.e. is this is a summary across different datasets/backbones?
- Calibration is not evaluated on OOD shifts. Networks are also evaluated without an off-the-shelf post-hoc calibration method which would be fairly easy for any practitioner to implement.
- Simplicity of design and ease of implementation / flexibility are important factors that survey could cover more comprehensively. Parameter count does not necessarily equate to simplicity. For example, for ease of implementation, we might document how many open source frameworks the detectors are implemented in.

The strengths of the paper outweigh the relatively minor weaknesses.

---

> ### Author Response · Authors · 2022-07-10
> **Response to Reviewer CDPx**
>
> Thank you for your feedback. We have addressed and included the requested changes in the revised submission (changes in blue).
> - We added the implementation details of Figure1 in Section A.1 in the Appendix and changed the caption for more clarification. We also extended the figure to now include both robustness to natural corruptions and adversarial robustness to provide a more holistic overview.
> In Addition, we added a new section (Section 12) to evaluate the adversarial robustness of all detectors in the revised paper.
> - To include a calibration study on an OOD dataset, we performed the analysis on the corrupted COCO dataset. As this exhibits a pattern similar to the IID case, we added Figure 21 in the Appendix.
> - Simplicity of design and ease of implementation would help the community at large. We added this new section (Section A.2) in the Appendix where we describe the problems we faced in reproducing individual results, the quantity and/or quality of repositories in different frameworks, and the ease of converting the networks to run on hardware devices.

---

> > ### Comment · Reviewer_CDPx · 2022-07-27
> > **Response**
> >
> > Thanks, the response and updated paper addresses some of my main concerns.

---

> > > ### Author Response · Authors · 2022-07-28
> > > **Response to Reviewer CDPx**
> > >
> > > Thank you for your response. Should there be any concern that wasn't addressed, we would be happy to do so. Kindly let us know if you need more information.

---

### Review · Reviewer_9pJM · 2022-06-28

**Summary Of Contributions:**

This paper is a survey of modern object detection approaches with a focus on real-time architectures.  The authors (to quite an impressive extent) compare architectures in an apples-to-apples fashion and going beyond some previous surveys evaluate on not just COCO and Pascal VOC, but a number of other domains (self driving, healthcare).  Moreover the authors focus on metrics beyond speed and accuracy, including quantities like energy consumption, robustness and reliability, as well as benchmarks on edge devices. Overall the paper should serve as a reasonably good handbook for a practitioner who needs to select an architecture.


**Requested Changes:**

The following are recommendations that would strengthen the work (though not critical to securing my recommendation for acceptance).

* *Correlations/dependencies between metrics*: a number of metrics are correlated with each other — this deserves a longer discussion.  For example, what causes a model to do well on energy consumption than can be expected by its FLOP count?
* *Self-contained explanations of detectors*: Section 3 is a nice survey of detectors from the last several years - however some of the descriptions are not completely self contained (the overall mechanism of the detectors could not be guessed based on description).  One example: Eqn 6: how does this equation capture classification into K categories at the end?  I recommend having a reader who is familiar with classifiers but not object detectors try reading this section and then try explaining the training procedure to the authors.
* *Other ideas for out-of-distribution generalization*: Another idea would be to train on some cities then test on hold-out cities (see, e.g., Google’s Auto-Arborist dataset).  Or train on static images and test on video frames.  (Note this is just a suggestion for strengthening the work, not a condition of acceptance)
* *CenterNet vs FCOS*: I’m picking out this particular pairing (but I think the comparison might be relevant to other pairs in the work too).   The original CenterNet best results were achieved with an hourglass backbone.  Both CornerNet and CenterNet showed results that (for whatever reason), they were able to achieve much stronger with hourglass shaped backbones than with standard ResNet with FPN backbones.  So I’m wondering if we are being fair to CenterNet in this paper since when we look at mean AP from both papers, the performance of the two architectures are not so different, but according to this paper FCOS is flat-out superior to CenterNet.
* *Batch norm folding*: Do the authors fold batch-norm into convolutions when benchmarking?
* *Non max suppression details*: Noting the effect of the confidence threshold on speed is nice — could the authors explain this?  Is the extra running time due purely to NMS? If so, I think this issue could be discussed more carefully. In practice, we have to choose an operating point and since we don’t always know what choice will be made, one way to view the mAP metric is as an integrated metric over all choices of operating point. Thus a very low threshold (e.g. even 1e-8) makes sense for computing mAP.  But of course in practice, we’d often use a much higher threshold which would reduce the NMS portion of compute.  Thus setting a confidence threshold of 0.4 is more about achieving a particular precision/recall tradeoff (and we might not discuss mean AP in this setting since it is the average over all choices of confidence threshold). Note also that running time might depend on NMS variant (some implementations of soft NMS can be much slower! — so what is the running time complexity of the implementation used in this work?)
* *Robustness*: I recommend comparing results here to those in “Accuracy on the Line: on the Strong Correlation Between Out-of-Distribution and In-Distribution Generalization” by Miller et al.  Specifically if you look at Figure 16, a striking feature of these plots is the relative ordering among the different detectors remains mostly unchanged over all corruption levels.  This meshes well with the result in Miller et al.  And it is not surprising therefore that FCOS is the most robust detector (given that it performs the best under no corruption).
* *Calibration*: Could calibration not be fixed to a large extent by applying something simple like platt scaling?
* *Misc fixes/questions (apologies that I wrote these out of order)*:
    * Can you plot speed accuracy tradeoff?  (I.e., speed on one axis, mean AP on the other  — as is done by many prior works).  This would allow us to see a “pareto-optimal” front of detectors.
    * Is Figure 1 averaged over (e.g. resolution, backbone, dataset etc?) Are scales comparable?  It seems the variation in one metric might be very small but variation in another might be very large…
    * Eqn 7: Lconf is not defined
    * Fig 4: typo: enchancement
    * Also everything is written out horizontally except NMS which looks funny
    * Clarification: In Sec 9.3 - what does it mean to modify anchor dimensions using a Gaussian?
    * Usage of “quintessentially” in Xception paragraph in Section 6 — is this correct?
    * Paragraph above Eqn 14 (second paragraph of Section 5.8) mentions that FCOS and CenterNet solve the overlapping issue by using centerness and FPN (just note that CenterNet doesn’t use FPN - or else there is also a problem with Table 2)
    * Part of table 4 shows up in a larger font
    * The colors in Figure 15 are somewhat confusing — I think the dark green is meant to indicate that blue goes all the way to the top and then the gap is the difference between the “blue” (which we can’t actually see) and the red dotted line.  It might be more intuitive to visualize the gap with an arrow


**Strengths And Weaknesses:**

### Strengths:

This survey contains good coverage of many modern architectures and some of the specific focuses in this work (e.g. real time inference, out-of-distribution performance) separate it from prior object detection surveys.  Moreover apples-to-apples comparisons of these approaches (done quite well in this work) is often very difficult to do — typically numbers are just cited from prior papers with little effort to standardize on backbones or hardware. One surprising outcome for example is that older architectures like SSD are shown to still be somewhat competitive when evaluated using comparable backbones that today’s architectures use.

### Weaknesses:

On the other hand, it’s worth at least discussing some of the apples-to-apples choices more carefully.  For example, it’s not completely obvious that the “right” thing to do is to put all models on the same footing during training (e.g. training with batch size 32).  For example, what if some model for whatever reason really needs a large batch size?  Alternatively, we could let the performance of a model be defined by the optimum performance it can reach with a given input resolution and trained on consistent hardware…

Also, as far as I can tell, there are no details given about the specific implementations used of each architecture here.  Did the authors reimplement everything? If so, how did they verify that the were able to capture the original authors’ implementation faithfully?  And if they reimplemented everything, will there be a code release?  (This point definitely should be addressed if it is to be accepted).

Finally, a number of followup works are not addressed in this paper (e.g. further versions of MobileNet and Yolo — e.g. Yolo v5, mobilenet v3 — and these choices to omit certain architectures should be explained)

---

> ### Author Response · Authors · 2022-07-10
> **Response to Reviewer 9pJM**
>
> **Response to requested changes:**
> -  To shed more light on the correlation between parameters, we added two more relevant graphs (Figure 13 and Figure 14)  in the ‘Detailed Analysis’ section of Results (Section8). The first shows the speed-accuracy trade-off for all heads and backbones. The second shows the correlation between different metrics. Accordingly, we discussed this in the revised paper. Hope this adds more clarity.
> - We took the feedback into account and added some more explanation to Sections 3, 4, and 5.1.
> - Thanks for the idea and the new datasets. They would enhance the OOD study but currently, it is computationally expensive to perform it in the given time frame (training 8heads + 9 backbones).  Also, as these datasets do not have standard usage in literature, a hyperparameter search is required to find the best settings that work for them. We currently have two OOD evaluations: on natural corruption (Corrupted COCO) and, on a different AD dataset (Cityscapes). Additionally, we have now included a new study on adversarial robustness. This is reported in Section 12 of the revised paper.
> - CenterNet-Hourglass does have better performance (and we reproduced this when we originally reimplemented CenterNet). But as we wanted to focus on the real-time performance of networks, we do not consider this backbone for the paper. In the real-time setting, CenterNet still offers a better trade-off between accuracy and speed compared to FCOS and is also the most robust against adversarial attacks (Figure 19).
> - We do not use batch norm folding in our experiments and we also added this detail in our experimental setup to avoid any ambiguity. However, the TRT conversion combines multiple layers in the network (including convolution and batch normalization layers) to enable efficient inference.
> - Thanks for pointing out this section. To clarify, there are two thresholds: confidence threshold and NMS threshold.  The confidence threshold is used to filter out the low confidence detections before sending it to the NMS module and then for evaluation. The NMS threshold is used to filter out the redundant overlapping predictions. NMS threshold is always a constant in standard literature and is set to 0.45. Whereas the confidence threshold used to filter low confidence thresholds is not usually mentioned in the literature. And this plays an important role in both accuracy and speed, a lower threshold ensures more boxes, and the precision increases. But for applications, the confidence threshold is set higher to remove noisy predictions.
> - This is an interesting paper and matches the trend in our analysis and we cite this in our results section (Section 11).
> - Multiple fixes exist for fixing calibration and we added this discussion at the end of Section 10. Several calibration solutions existing for classification (such as Platt Scaling, and temperature scaling) could also be applied for detection. But our focus is more on evaluating and comparing the reliability of the networks as is, and hence do not apply any of the solutions to alleviate miscalibration.
> - addressing the misc fixes:
>   - We added the speed-accuracy tradeoff plot (Figure 13) which will provide more clarity.  We also explained the implementation details of Figure1 in Section A.1 in the Appendix.  Figure 1 is modified and now includes two additional metrics: natural robustness and adversarial robustness.
>   - Figure 1 implementation details are now added in Section A.1.
>   - Equation term definition is added.
>   - Typos are fixed.
>   - Schematic diagram is now fixed.
>   - Added some more explanations about modifying anchor dimensions. Please let us know if it is clear.
>   - That word is out of place, we removed it.
>   - Thanks for bringing this to our attention. It was an ambiguous statement, we wanted to convey the centerness of both networks along with FPN of FCOS. But we removed the sentence to avoid any confusion.
>   - For calibration plot, *“dark green is meant to indicate that blue goes all the way to the top and then the gap is the difference between the “blue” (which we can’t actually see) and the red dotted line”* - this is indeed correct. We have added stripes to all the (three) calibration plots, to make it more clear. Hope the new plots are better.

---

> ### Author Response · Authors · 2022-07-10
> **Response to Reviewer 9pJM**
>
> Thank you for your feedback. We would like to address the concerns and the requested changes below.
>
> **Responses to weaknesses:**
> - Comparison of multiple networks on a level field is indeed tricky. The original works of all detectors do report the best accuracy of that network in the best parameter setting. Sometimes they require higher batch sizes, bigger resolutions, or heavier backbones to achieve the optimum performance. But this makes it difficult to compare between multiple networks. Hence, we opted for the setting that can be fair to all the methods included in our paper, for instance, if a method needs a longer training to converge, we used that training budget for all the other methods, or if one method depends on pretraining, we used the pre-trained weights for the rest, too. That way, we provide a platform to compare them on a uniform setting, but by keeping the core properties still the same as their original settings (such as using ADAM optimizer for transformers, lower learning rate for TTFNet, etc).
> - Thanks for bringing this point. We address this here and also clarify the same in the experimental section (Section 7.3)
>   - We did indeed reimplement all the architectures. The procedure we followed was: first, reimplement all the networks (backbones and heads) in our repository. We have a PyTorch framework with all the architectures implemented in it. Second, reproduce all the original results with the settings provided in the paper (we struggled in some cases to reproduce due to missing code or parameters (refer to Section A.2). Third, train all the networks using the standard evaluation settings and report the results.
>   - We would like to release our repository for research purposes, but it would require some internal discussions as this project is privately funded. Hence, the decision is out of our hands.
> - Choosing the networks was a tough decision as multiple networks are getting introduced frequently. Multiple versions of YOLO detectors keep getting released. The improvements have not been due to a significant change at the architectural level. YOLOv4 relies on several bags of freebies and specials to supersede its successors. The improvements in YOLOv5 include a new mosaic data augmentation and auto-learning bounding boxes. Both perform well on the heavier CSPDarknet53 backbone. We chose to draw the line with one version of YOLO and concentrate more on the different types of detectors (anchor-based, keypoint-based, and transformer-based). Also, another reason was the difficulty in reproducing YOLO results (the official implementation of many YOLO versions is in C and we consistently struggled in reproducing the results of the papers).

---

### Decision · Action_Editors · 2022-07-29

**Recommendation:** Accept as is

**Comment:**

The problem of object detection is one of the most important applications in computer vision requiring not only the ability to identify not just what resides in an image but also where the object instance resides. In the last few years there have been a multitudes of advancements in the field of object detection including basic backbone architectures, detection stems, cascade architectures, data augmentations, etc. Stitching together what all of these individual advancements mean for the larger field or a modern day practitioner is a difficult enterprise because it is difficult to ascertain what methods are additive and what trade-offs result from employing various methods in tandem on modern detection problems.

This paper provides a survey of many of the most important modern object detection approaches with a focus on real-time architectures. The authors go to great lengths of provide an in-depth analysis of the accuracy, energy consumption, robustness, calibration, and speed. They test these methods not just on standard academic datasets (e.g. Pascal VOC and COCO) but on non-standard and important problems in medical imaging and self-driving applications and provide more comprehensive evaluations of real-time performance. The reviewers highlight how this work provide a great handbook for future practitioners in the field. Given the many advancements in the field, this survey additionally provides a common evaluation framework and benchmarks for future advancements in the field. The paper is accepted provided assuming all of the comments from the reviewers are fully addressed.

---

> ### Author Response · Authors · 2022-08-03
> **Camera-ready**
>
> Dear Jonathon and reviewers,
>
> We would like to thank you for your suggestions,  valuable feedback, and encouraging words. We have incorporated all the requested changes and submitted the new PDF.
>
> Authors